# The Harder Path: Last Iterate Convergence for Uncoupled Learning in Zero-Sum Games with Bandit Feedback

Côme Fiegel [1 2]  Pierre Menard [3]  Tadashi Kozuno [4]  Michal Valko [5]  Vianney Perchet [1 2 6]

## Abstract

We study the problem of learning in zero-sum matrix games with repeated play and bandit feedback. Specifically, we focus on developing uncoupled algorithms that guarantee, without communication between players, the convergence of the last-iterate to a Nash equilibrium. Although the non-bandit case has been studied extensively, this setting has only been explored recently, with a bound of $\mathcal{O}(T^{-1/8})$ on the exploitability gap. We show that, for uncoupled algorithms, guaranteeing convergence of the policy profiles to a Nash equilibrium is detrimental to the performance, with the best attainable rate being $\Omega(T^{-1/4})$ in contrast to the usual $\Omega(T^{-1/2})$ rate for convergence of the average iterates. We then propose two algorithms that achieve this optimal rate up to constant and logarithmic factors. The first algorithm leverages a straightforward trade-off between exploration and exploitation, while the second employs a regularization technique based on a two-step mirror descent approach.

## 1. Introduction

In *zero sum matrix games*, two players each take a single action and accordingly receive for the first player a loss and the second player a gain of the same magnitude. Such games always admit for each player at least one minimax policy (a specific case of Nash equilibrium), assuming a stochastic choice of action (v. Neumann, 1928). Computing these policies, however, is non-trivial and requires knowledge of the underlying game matrix of payoffs.

We are interested in learning minimax policies by repeat-

edly playing the game. Specifically, we consider the *bandit feedback* setting, where the payoff matrix is unknown and players only observe their losses or rewards.

Within the context of *uncoupled learning*, each player learns the minimax policy independently, without communication or knowledge of their opponent's actions. This setup motivates the use of methods similar to those employed in classical single-player online learning (Cesa-Bianchi & Lugosi, 2006), where a player adapts to play optimally in an adversarially changing environment.

These methods usually bound the player's regret, defined as the difference between the best possible cumulative loss under a fixed policy and the actual cumulative loss incurred by the player. They are known to be applicable in the context of games to compute minimax policies (Cesa-Bianchi & Lugosi, 2006). They however have two well-known weaknesses:

- The policies played over the iterations generally do not converge or even approach an equilibrium.

- Instead, the average policy is computed and outputted as a proxy for convergence. While averaging tabular policies is simple and inexpensive, this step is not as simple for practical applications. For instance, it is not clear how to compute the average of policies represented by a neural network (Heinrich et al., 2015; McAleer et al., 2022).

In part for this reason, a significant portion of the recent literature studies methods with *last-iterate convergence* for which the actual policies played over time converge toward a Nash equilibrium. Some algorithms, such as *Optimistic Mirror Descent* (OMD, Popov 1980; Rakhlin & Sridharan 2013), exhibit this convergence despite being initially proposed for their regret-bounding properties, but with a vastly different analysis. For the OMD algorithm, in the deterministic *full-information* feedback setting, the rate of convergence of the exploitabilty gap toward zero is even improved, transitioning from $\mathcal{O}(1/T)$ for the average to a linear rate for the last iterate (Wei et al., 2021).

However, the literature on last-iterate convergence with stochastic feedback is limited, and even more so when con-

---

*Equal contribution [1]ENSAE Paris - CREST, Palaiseau, France [2]Inria - FairPlay [3]ENS Lyon, Lyon, France [4]OMRON SINIC X, Tokyo, Japan [5]Stealth AI Startup / Inria / ENS [6]Criteo AI Lab, Paris, France. Correspondence to: Côme Fiegel <come.fiegel@normalesup.org>.

*Proceedings of the 42nd International Conference on Machine Learning*, Vancouver, Canada. PMLR 267, 2025. Copyright 2025 by the author(s).

sidering bandit feedback. While properties of the last-iterate convergence have been studied in the broader context of stochastic variational inequalities, these works often rely on assumptions that are not applicable to matrix games, such as second-order sufficiency (Azizian et al., 2021), in addition to not accounting for the bandit feedback aspect.

Recently, some methods (Cai et al., 2023; Dong et al., 2024) were proposed for this specific problem. They however only obtained an upper bound of $\mathcal{O}(T^{-1/8})$ on the exploitability gap with high probability [1]. Considering the best known lower bound was $\Omega(T^{-1/2})$ from a reduction to the $K$-arms bandit problem, Cai et al. (2023) raised the question of whether a better lower bound was achievable.

In this work, we address the following question: *What is the best attainable rate for last-iterate convergence of uncoupled learning with bandit feedback in matrix games?*

## 2. Contributions

We focus in the above settings on zero-sum matrix games.

- We provide a better lower bound for the problem of learning a minimax profile with uncoupled convergent algorithms and bandit feedback. This lower bound, of $\Omega(T^{-1/4})$ for the $L^p$ convergence given $p \in [2, \infty]$, shows that guaranteeing last-iterate convergence is harder than just guaranteeing convergence for the average iterates, which can be done at a rate $\mathcal{O}(T^{-1/2})$.

  Intuitively, this relies on the fact that, at least for some simple $2 \times 2$ games, there exists a minimax policy for one player that renders all actions of the other player equivalent, thereby preventing any learning on their part. Meanwhile, a policy converging to this minimax policy may not entirely prevent learning but will significantly slow it down, as the difference between the two action rewards converges to zero.

- This lower bound is also stated for $p \in (0, 2]$, for which it improves to $\Omega(T^{-1/(2+p)})$.

- We propose a general simple framework for transforming an algorithm with classical anytime guarantees into one with last-iterate guarantees, based on a simple exploration-exploitation trade-off. We use this framework to show that the above lower bound is tight: with the `EXP3-IX` of Kocák et al. (2014), a $\widetilde{\mathcal{O}}(T^{-1/(2+p)})$ rate can be attained for the $L^p$ convergence, given any $p$ in $(0, 2]$. However, the computation of some average policies is still needed, as the framework relies on

the output of the underlying algorithm, which is here regret-based.

- We also propose a more practical algorithm, based on a strong regularization of the problem, which does not require the computation of an average policy. It enjoys a $\mathcal{O}(T^{-1/4})$ rate, thanks to the use of an unbiased estimate of the losses in contrast to previous approaches. However, even if the last iterate converges, the algorithm is not completely anytime as the regularization must be chosen with the knowledge of the horizon $T$. To address this limitation, a doubling trick approach is stated, which features the same exploration-exploitation trade-off.

The different rates are summarized in Table 1.

## 3. Related works

**Variational inequalities**  The problem of finding a Nash equilibrium for a matrix game can be formulated as finding the solution of a specific Lipschitz variational inequality (Mancino & Stampacchia, 1972), which is furthermore monotone when the game is zero-sum.

Finding algorithms for solving monotone variational inequalities is a major part of the optimization literature, starting with the proximal point algorithm (Martinet, 1970; Tyrrell, 1976). In particular, using two samples at each iteration, the Extra-Gradient method (Korpelevich, 1976; Nemirovski, 2004) has a $\mathcal{O}(1/T)$ convergence for the average iterate. With a strongly monotone operator, the rate is even linear (Facchinei & Pang, 2004) for the last iterate.

For stochastic variational inequality, in which only an unbiased estimate of the operator is observed, a rate $\mathcal{O}(1/\sqrt{T})$ is obtained by Juditsky et al. (2011) for the average iterates. For strongly monotone operators, this rate can be improved to $\mathcal{O}(1/T)$ (Nemirovski et al., 2009).

An important way to generalize some of the aforementioned methods is through the *mirror descent* approach, which extends Euclidean methods to other geometries, as discussed by Nemirovskiĭ & IUdin (1983). This generalization is particularly crucial when dealing with bandit feedback, with the `EXP3` algorithm, introduced by Auer et al. (2002), serving as a key example of this approach under the Kullback-Leibler geometry.

Instead of relying on two samples at each iteration as in the Extra-gradient method, the *Optimistic Mirror Descent* reuses the previous one as an estimate. While it can be traced back to Popov (1980), it has regained interest relatively recently (Chiang et al., 2012; Rakhlin & Sridharan, 2013).

---

[1] Dong et al. (2024) obtained a rate $\mathcal{O}(T^{-1/4})$ on the Kullback-Leibler divergence between the profile and the Nash equilibrium, which only translates in general into a rate $\mathcal{O}(T^{-1/8})$ for the exploitability gap

| Algorithm | Convergence | Rate |
|---|---|---|
| Cai et al. 2023 (Algorithm 1) | *High probability* | $\widetilde{\mathcal{O}}(T^{-1/8})$ |
| | $L^2$ | $\widetilde{\mathcal{O}}(T^{-1/6})$ |
| Dong et al. 2024 (Algorithm 1) | *High probability* | $\widetilde{\mathcal{O}}(T^{-1/8})$ |
| Simultaneous Explore or Exploit (this paper) | $L^p, p \in (0, 2]$ | $\widetilde{\mathcal{O}}(T^{-1/(2+p)})$ |
| Uncoupled Regularized EXP3 (this paper) | $L^2$ | $\widetilde{\mathcal{O}}(T^{-1/4})$ |
| Lower bound (this paper) | $L^p, p \in (0, 2]$ | $\mathcal{O}(T^{-1/(2+p)})$ |

*Table 1.* Rate of convergence of the exploitability gap of algorithms with last-iterate guarantees under bandit feedback. The notation $\widetilde{\mathcal{O}}$ hides logarithmic dependences in $T$, and in $\delta$ for algorithms that converges with probability at least $1 - \delta$.

**Learning in games** In the case of a *deterministic* feedback, classical regret-bounding algorithms can be shown to enjoy a $\mathcal{O}(1/\sqrt{T})$ rate for the average iterates. Optimistic mirror descent in particular improves this rate to $\mathcal{O}(1/T)$ (Rakhlin & Sridharan, 2013; Kangarshahi et al., 2018). Using a problem-dependent constant, the rate can even be shown to be linear (Tseng, 1995; Wei et al., 2021) for the last iterate. However, Cai et al. (2024) recently proved that for some algorithms, including OMD, obtaining a $\mathcal{O}(1/T)$ rate for the last iterate without this problem-dependent constant is impossible.

This work focuses on learning a minimax strategy with *bandit feedback* using uncoupled algorithms. In the context of smooth monotone games, several recent studies, building on the foundational work of Bravo et al. (2018), have explored a setting where only the value associated with the chosen policy profile is observed (Hsieh et al., 2019; Drusvyatskiy et al., 2022; Tatarenko & Kamgarpour, 2022; Huang & Hu, 2024; Ba et al., 2025). However, this approach does not account for the inherent stochasticity present in the $K$-arms bandit problem:: particularly in matrix games, this observed value represents the average of the rewards under the policies, rather than the reward of a single sampled action profile.

Meanwhile, Abe et al. (2023) considered a stochastic feedback, but without the specific bandit aspect. Muthukumar et al. (2020) showed an impossibility result that some of the algorithms with no-regret guarantees cannot converge almost surely to a Nash equilibrium.

In our setting, Cai et al. (2023) recently showed that a $\mathcal{O}(T^{-1/8})$ rate can be attained, a result later extended by Dong et al. (2024) to smooth monotone games.

## 4. Setting

**Zero-sum matrix game** Two players, called the min- and the max-player, respectively play actions $a \in \mathcal{A}$ and $b \in \mathcal{B}$, in sets of cardinality $A$ and $B$, to receive a loss $L(a, b) \in [0, 1]$: the min-player wants to minimize this loss, while the max-player wants to maximize it.

The two players are allowed to play stochastically: they choose two mixed policies $\mu \in \Delta_A := \left\{ \mu, \sum_{a=1}^{A} \mu(a) = 1 \right\}$ and $\nu \in \Delta_B := \left\{ \nu, \sum_{b=1}^{B} \nu(b) = 1 \right\}$ and optimize their choice according to the expected loss $L(\mu, \nu)$ defined by:

$$L(\mu, \nu) = \mathbb{E}_{a \sim \mu, b \sim \nu} \left[ L(a, b) \right] .$$

A tuple $(\mu, \nu) \in \Delta_A \times \Delta_B$ of policies will be called a profile

We look to obtain a minimax profile (a special case of the later Nash-equilibrium, Nash Jr 1950), defined as a profile $(\mu^\star, \nu^\star)$ that satisfies

$$\mu^\star \in \underset{\mu^\dagger \in \Delta_A}{\arg\min} \, L(\mu^\dagger, \nu^\star) \quad \text{and} \quad \nu^\star \in \underset{\nu^\dagger \in \Delta_B}{\arg\max} \, L(\mu^\star, \nu^\dagger),$$

whose existence is guaranteed (v. Neumann, 1928).

The proximity of a profile $(\mu, \nu)$ to the set of Nash equilibria can be characterized using the exploitability gap:

$$EG(\mu, \nu) = - \min_{\mu^\dagger \in \Delta_A} L(\mu^\dagger, \nu) + \max_{\nu^\dagger \in \Delta_B} L(\mu, \nu^\dagger)$$

Note that the exploitability gap is zero if and only if $(\mu, \nu)$ is a Nash equilibrium.

**Sequential learning with bandit feedback** We assume that at each iteration $t$, both players select some policies $\mu^t$ and $\nu^t$, sample two actions $a^t \sim \mu^t$ and $b^t \sim \nu^t$, and get a stochastic loss $\ell^t \in [0, 1]$ associated to these two moves. Formally, we assume the existence, for each $a \in \mathcal{A}$ and $b \in \mathcal{B}$, of a probability distribution $p(a, b)$ on $[0, 1]$ such that

$$\forall t \in \mathbb{N}, \, \ell^t | \mathcal{F}^{t-1}, a^t, b^t \sim p(a^t, b^t)$$
$$\text{and} \quad \mathbb{E}_{\ell \sim p(a,b)} [\ell] = L(a, b)$$

where $\mathcal{F} = (\mathcal{F}^t)_{t \in \mathbb{N}}$ is a filtration recursively defined by the observations,

$$\mathcal{F}^t = \sigma \left( \omega, a^1, b^1, \ell^1 ..., a^t, b^t, \ell^t \right) ,$$

with $\omega$ the internal randomness (extra to the sampling of actions) of both players.

This filtration summarizes all information available to both players up to the beginning of round $t + 1$. A sequence of profile $(\mu^t, \nu^t)_{t \in \mathbb{N}}$ will especially be called a *learning sequence* if it is predictable with respect to $\mathcal{F}$.

**Last-iterate convergence**   We define, for all $p > 0$ the $F$-norm $\|\cdot\|_p$ (Banach, 1932) for some real random variable $X$ by

$$\|X\|_p := \mathbb{E}\left[|X|^p\right]^{\frac{1}{p}}.$$

Now, let $f = (f^t)_{t \in \mathbb{N}}$ be a positive sequence decreasing to 0. A learning sequence $(\mu^t, \nu^t)$ will satisfy a $L^p$ last-iterate convergence of rate $f$ if

$$\forall t \in \mathbb{N}, \|EG(\mu^t, \nu^t)\|_p \leq f^t.$$

It is asymptotic if there simply exists some $T_0 \in \mathbb{N}$ such that

$$\forall t \geq T_0, \|EG(\mu^t, \nu^t)\|_p \leq f^t.$$

This requires both players to play policies they deem near-optimal during each iteration, instead of simply outputting one good policy each at the end of the procedure.

**Output convergence**   On the other end, an algorithm has an $L^p$ *output convergence* of rate $g$, with $g = (g^t)_{t \in \mathbb{N}}$ a sequence decreasing to 0, if it can output a learning sequence $(\hat{\mu}^t, \hat{\nu}^t)$ such that:

$$\forall t \in \mathbb{N}, \|EG(\hat{\mu}^t, \hat{\nu}^t)\|_p \leq g^t.$$

This restriction is weaker than the previous last-iterate convergence, as it does not put any constraint on the actual policies played at each round. It however still forces the algorithm to have anytime guarantees.

**Uncoupled algorithm**   We say an algorithm is *uncoupled* if it independently controls the two players, without active communication between the two instances. In particular, we assume that observation of the opponent's action is not possible.

This definition is a bit informal as completely characterizing the impossibility of communication is hard mathematically. Indeed, even if direct communication is forbidden, the two instances are not isolated as the loss of one player still depends on the actions of the other. Nothing technically prevents one player from passing bits of information to the other through artificial choices of policies.

## 5. Lower bound

In this section, we establish that these assumptions combined are quite restrictive. Especially, we show it is impossible to guarantee a better rate than $\otimes(T^{-1/(2+p)})$ for the $L^p$ last-iterate convergence with $p \in (0, 2]$.

For this purpose, we will consider the following $2 \times 2$ games, for any $\varepsilon \in [-1/12, 1/12]$:

$$M^\varepsilon = \begin{bmatrix} \mathcal{B}(2/3 - \varepsilon) & \mathcal{B}(1/3 + \varepsilon) \\ \mathcal{B}(1/3) & \mathcal{B}(2/3) \end{bmatrix}$$

where $\mathcal{B}$ denotes a Bernoulli distribution.

For these games, regardless of the choice of $\varepsilon$ in the domain, it is easily shown that

$$\nu^\star = \begin{bmatrix} 1/2 \\ 1/2 \end{bmatrix}$$

is the only min-max max player policy, with an associated value of $1/2$. Furthermore, the exploitability gap of any profile $(\mu^t, \nu^t)$ is at least proportional to the distance $|\delta^t|$ between $\nu^t$ and $\nu^\star$, with $\delta^t = \nu^t(1) - 1/2$.

On the contrary, the min-max min-player policy depends on the choice of $\varepsilon$, as there exists no policy for the min-player policy that is good regardless of this choice. Especially, the exploitability gap of any profile $(\mu, \nu)$ can be shown to be at least proportional to $|\varepsilon|$ for one of the two games $M^{-\varepsilon}$ and $M^\varepsilon$.

We therefore assume that the game matrix is either $M^\varepsilon$ or $M^{-\varepsilon}$, but that the exact choice is unknown. At each iteration $t$, conditioning on $\mathcal{F}^{t-1}$, the law of the reward vector (i.e. the law of the rewards for each action) for the min-player is given by:

$$\text{with } M^\varepsilon : \begin{bmatrix} \mathcal{B}\left(1/2 + \delta^t/3 - 2\delta^t\varepsilon\right) \\ \mathcal{B}\left(1/2 - \delta^t/3\right) \end{bmatrix}$$
$$\text{with } M^{-\varepsilon} : \begin{bmatrix} \mathcal{B}\left(1/2 + \delta^t/3 + 2\delta^t\varepsilon\right) \\ \mathcal{B}\left(1/2 - \delta^t/3\right) \end{bmatrix},$$

where we re-used the value $\delta^t$ characterizing the difference between $\nu^t$ and the min-max policy $\nu^\star$.

Guaranteeing an exploitability gap in $o(\varepsilon)$ at any horizon $T$ implies the min-player can discriminate between these two options. Using some additive properties of the Kullback-Leibler divergence, we know that, without the observation of the max-player actions $(b^t)_{t \in \mathbb{N}}$, this discrimination is only possible with an arbitrarily high probability when:

$$\sum_{t=1}^{T} \mu^t(1)(\delta^t\varepsilon)^2 \, \mathbb{I}_{\{a^t=1\}} = \Omega(1).$$

Given full control of $\delta^t$ over the iterations, the ideal choice would be $\delta^t = \pm 1/2$, which gives the usual lower bound

$|\varepsilon| \geq \Omega(T^{-1/2})$. However, the last-iterate convergence assumption implies that $\delta^t$ must also converge to 0. Depending on how strong this convergence of $\delta^t$ must be, the minimum value of $|\varepsilon|$ that allows the discrimination greatly increases.

This idea is formalized in the following theorem, proven in Appendix A.

**Theorem 5.1.** *Assume that the sampled action $b^t \sim \nu^t$ of the max-player at each iteration $t$ is never observed. Then, for any $p \in (0, 2]$, no learning sequence can achieve simultaneously for all $2 \times 2$ matrix games described above an $L^p$ last iterate convergence of*

$$f^t = \frac{4^{\frac{-1}{p}}}{118} t^{\frac{-1}{2+p}},$$

*even asymptotically.*

*Remark* 5.2. As mentioned in Section 4, the uncoupling of the two instances is hard to formalize mathematically. In the above theorem, this formalization is done through the assumption that the max player's sampled action is never observed. As the max-player already knows its optimal policy $\nu^\star$ and consequently does not need this knowledge for learning, we consider this assumption to be reasonable.

Furthermore, this assumption of not observing the opponent's moves is important for the rate. Indeed, the observation of these moves would allow the estimation of each entry of the game matrix at a rate $\widetilde{\mathcal{O}}\left(t^{-1/2}\right)$, assuming that each action is played for a non-negligible proportion of the iterations. With this estimated game matrix, asymptotically playing the associated minimax profile would lead to the same $\widetilde{\mathcal{O}}\left(t^{-1/2}\right)$ rate for the exploitability gap.

**The p > 2 case:** Theorem 5.1 only deals with $p \in (0, 2)$. For $p \in (2, \infty]$, using $\|\cdot\|_p \leq \|\cdot\|_2$ for any random variable (Rudin, 2006), we immediately get the following corollary.

**Corollary 5.3.** *Under the same assumption as Theorem 5.1, for any $p \in (2, \infty]$, no learning sequence can achieve simultaneously for all $2 \times 2$ matrix games described above an $L^p$ last iterate convergence of*

$$f^t = \frac{1}{236} t^{\frac{-1}{4}},$$

*even asymptotically.*

The next section shows that the lower bound cannot be improved in rate for $p \in (0, 2]$, as it is tight up to some constant and logarithmic factors. We conjecture that, despite the above loose bounding, it also cannot be improved for $p \in (2, \infty)$.

## 6. Exploration-Exploitation trade-off

The above theorem shows that for uncoupled algorithms, $L^p$ last-iterate convergence is strictly harder than output con-

---

**Algorithm 1** Simultaneous Explore or Exploit

1: **Input:** Algorithm $\mathcal{A}$ with $L^p$ output convergence
   Sequence $(p^t) \in [0, 1]$
2: **Initialize:** $k^0 \leftarrow 0$
   **Draw** $u \sim \mathcal{U}([0, 1])$ common to both players
3: **Algorithm:** For $t = 1$ to $+\infty$:
   $k^t \leftarrow \lfloor \sum_{i=1}^t p^i + u \rfloor$
   **If** $k^t > k^{t-1}$:
       **Play** current profile $(\mu_{\mathcal{A}}^{k^t}, \nu_{\mathcal{A}}^{k^t})$ and **update** algorithm $\mathcal{A}$
       **Otherwise**:
       **Play** the output $(\hat{\mu}_{\mathcal{A}}^{k^t}, \hat{\nu}_{\mathcal{A}}^{k^t})$
4: **Output:** Learning sequence of policies with $L^2$ last-iterate convergence

---

vergence. Indeed, the anytime version of `EXP3-IX` (Kocák et al., 2014; Neu, 2015) achieves a rate of $\widetilde{\mathcal{O}}(t^{-1/2})$ for the average profile with high probability, which can be translated into the same rate for the $L^p$ output convergence for any $p > 0$.

While these two ways of converging are not equivalent, there is a procedure that transforms any algorithm $\mathcal{A}$ with $L^p$ output convergence guarantees into one with $L^p$ last-iterate convergence, at a price of a worse rate. The idea is simple: at each iteration, either both players "explore" by playing one iteration of $\mathcal{A}$ and updating accordingly, or "exploit" by both playing the current estimate.

If the probabilities $(p^t)_{t \in \mathbb{N}}$ of exploring at each iterations satisfy:

$$\lim_{t \to +\infty} p^t = 0 \quad \text{and} \quad \sum_{t=1}^{+\infty} p^t = +\infty,$$

the resulting learning sequence has last-iterate convergence. Indeed, the first condition makes the exploration contribution to the exploitability gap negligible asymptotically, while the second ensures that the outputs of $\mathcal{A}$ converge to some minimax policies.

The players need to be synchronized in their choices of exploration and exploitation steps. This can be done by sharing a single seed $u \sim \mathcal{U}([0, 1])$ as shown in Lemma B.1 of the appendix.

The procedure is summarized in Algorithm 1. The following lemma, proven in Appendix B, formalizes the above intuition, given some rate for the output of algorithm $\mathcal{A}$.

**Lemma 6.1.** *Assume that $\mathcal{A}$ satisfies an output convergence of rate $g$. Then Algorithm 1 satisfies an $L^p$ last iterate convergence of rate $f$ defined by*

$$f^t = 2^{\frac{1}{p}} \left[ (p^t)^{\frac{1}{p}} + g^{r^t} \right]$$

*where* $r^t = \left\lfloor \sum_{k=1}^{t} p^k \right\rfloor$.

We obtain the following theorem with the anytime version of EXP3-IX and the appropriate parameters.

**Theorem 6.2.** *Using Algorithm 1 with probabilities $p^t = t^{-p/(2+p)}$, and as $\mathcal{A}$, the algorithm EXP3-IX with parameters $\eta_{min}^t = 2\gamma_{min}^t = \sqrt{\log(A)/(At)}$ for the min-player and $\eta_{max}^t = 2\gamma_{max}^t = \sqrt{\log(B)/(Bt)}$ for the max-player, we obtain an $L^p$ last-iterate convergence rate of*

$$f^t = 17\sqrt{A+B}\, 2^{\frac{1}{p}} t^{-\frac{1}{2+p}} \log\left(4(A+B)t^2/p\right) .$$

This approach thus reaches the optimal rate $\widetilde{\mathcal{O}}(t^{\frac{-1}{2+p}})$ mentioned above for the $L^p$ convergence, up to some logarithmic and constant factors. However, it has several drawbacks that make it not applicable in real settings:

- No communication is required, but the players are not truly uncoupled in the usual sense as they still need to share a common seed (a sample from a uniform law at the beginning of the game). Note that this does not go against the hypothesis of Theorem 5.1, as this seed can be the random variable $\omega$ of the filtration $\mathcal{F}$.

- The exploitation steps are only performed to respect the anytime guarantees and have no practical use.

- One of the main points of the last-iterate convergence is to avoid the computation of the average necessary in the regret-based algorithm. Not only is this computation still required here (the output of EXP3-IX is an average), but also needs to be done at almost every iteration.

## 7. Regularized mirror descent

Because of these drawbacks, we propose another algorithm instead based on some regularized dynamics for the convergence.

**Loss estimation**  As explained in the settings, at each iteration $t$, the mean loss vectors $L(\cdot, \nu^t)$ and $1 - L(\mu^t, \cdot)$ for respectively the min and max players are not observed, only one sample $\ell^t$ is observed by both players. It can however be estimated through the importance sampling estimators:

$$\hat{\ell}_{min}^t = \frac{\hat{\ell}^t}{\mu^t(a^t)} \mathbb{I}_{\{a=a^t\}}$$

$$\hat{\ell}_{max}^t = \frac{1 - \hat{\ell}^t}{\nu^t(b^t)} \mathbb{I}_{\{b=b^t\}} .$$

While these estimators are unbiased, their variance is not bounded as the probabilities associated to any action can become arbitrarily small. For this reason, their use alone

theoretically prevents the convergence to the minimax policy, even in average (Kozuno et al., 2021). A simple way to counteract this issue is to use IX estimation (Neu, 2015) and add an additive term $\gamma^t$ to the denominator, but this biases the estimation and potentially worsens the rate. Theorem 7.2 shows that this bias is not necessary, at least for the $L^2$ convergence.

**Regularization**  Given $\tau > 0$, we define the regularized zero-sum game $L^\tau$ over $\Delta_A \times \Delta_B$ with

$$L^\tau(\mu, \nu) = \mathbb{E}_{\mu,\nu}\left[L(a,b)\right] + \tau \mathcal{D}_{KL}(\mu, \mu^0) - \tau \mathcal{D}_{KL}(\nu, \nu^0)$$

where $\mathcal{D}_{KL}$ is the Kullback-Leibler divergence between two distributions and $(\mu^0, \nu^0)$ an arbitrary profile. This game admits a unique Nash equilibrium, which will be denoted by $(\mu^{\star,\tau}, \nu^{\star,\tau})$.

Similarly to the regularization of a convex function in order to make it strongly convex, the point of this transformation is to transform the monotone *pseudo-gradient* operator $F : \Delta_A \times \Delta_B \to \mathbb{R}^{A+B}$ defined by $F(\mu, \nu) := (\nabla_\mu L(\mu, \nu), -\nabla_\nu L(\mu, \nu))$ into a strongly monotone operator $F^\tau(\mu, \nu) := (\nabla_\mu L^\tau(\mu, \nu), -\nabla_\nu L^\tau(\mu, \nu))$, which satisfies, for all $(\mu, \nu), (\mu', \nu') \in \Delta_A \times \Delta_B$:

$$\langle F^\tau(\mu, \nu) - F^\tau(\mu', \nu'), (\mu - \mu', \nu - \nu') \rangle$$
$$\geq \tau \left( \|\mu - \mu'\|_1^2 + \|\nu - \nu'\|_1^2 \right) .$$

A higher $\tau$ allows for faster convergence but at the price of a slightly different Nash-equilibrium $(\mu^{\star,\tau}, \nu^{\star,\tau})$. This implies a trade-off, as the exploitability gap of this regularized Nash equilibrium for the base game can be shown to be at most proportional to $\tau$.

**Regularized updates**  Given $\tau > 0$, a common way of updating is to do a mirror-descent with the regularized pseudo-gradient $F^\tau$, which is for example used by in this setting Cai et al. (2023). We instead use the following updates:

$$\mu^t = \underset{\mu \in \Delta_A}{\arg\min}(1 - \tau\eta^t)\mathcal{D}_{KL}(\mu, \mu^{\tau,t}) + \tau\eta^t \mathcal{D}_{KL}(\mu, \mu^0)$$

$$\nu^t = \underset{\nu \in \Delta_B}{\arg\min}(1 - \tau\eta^t)\mathcal{D}_{KL}(\nu, \nu^{\tau,t}) + \tau\eta^t \mathcal{D}_{KL}(\nu, \nu^0)$$

$$\mu^{\tau,t+1} = \underset{\mu \in \Delta_A}{\arg\min} \eta^t \left\langle \hat{\ell}_{min}^t, \mu \right\rangle + \mathcal{D}_{KL}(\mu, \mu^t)$$

$$\nu^{\tau,t+1} = \underset{\nu \in \Delta_B}{\arg\min} \eta^t \left\langle \hat{\ell}_{max}^t, \nu \right\rangle + \mathcal{D}_{KL}(\nu, \nu^t) .$$

This method is similar to the one proposed by Munos et al. (2023), but is here able to deal with non-symmetric games and bandit feedback. It works in two steps. First, the two

---

**Algorithm 2** Uncoupled Regularized EXP3

---

1: **Input:** Learning rates $\eta^t > 0$
   Regularization parameter $\tau > 0$
2: **Algorithm:** Initialize $\mu^{\tau,1}$ and $\nu^{\tau,1}$ to the uniform policies $\mu^0$ and $\nu^0$.
   **For** $t = 1$ to $+\infty$:
$$\mu^t \leftarrow \arg\min_{\mu \in \Delta_A} (1 - \tau\eta^t)\mathcal{D}_{\text{KL}}(\mu, \mu^{\tau,t}) + \tau\eta^t\mathcal{D}_{\text{KL}}(\mu, \mu^0)$$
$$\nu^t \leftarrow \arg\min_{\nu \in \Delta_B} (1 - \tau\eta^t)\mathcal{D}_{\text{KL}}(\nu, \nu^{\tau,t}) + \tau\eta^t\mathcal{D}_{\text{KL}}(\nu, \nu^0)$$
   **Sample and play** $a^t \sim \mu^t, b^t \sim \nu^t$
   **Observe** $\ell^t$
$$\mu^{\tau,t+1} \leftarrow \arg\min_{\mu \in \Delta_A} \eta^t \left\langle \hat{\ell}^t_{\min}, \mu \right\rangle + \mathcal{D}_{\text{KL}}(\mu, \mu^t)$$
$$\nu^{\tau,t+1} \leftarrow \arg\min_{\nu \in \Delta_B} \eta^t \left\langle \hat{\ell}^t_{\max}, \nu \right\rangle + \mathcal{D}_{\text{KL}}(\nu, \nu^t)$$
   where $\hat{\ell}^t_{\min} = \frac{\ell^t}{\mu^t(a^t)}\mathbb{I}_{\{a^t\}}$ and $\hat{\ell}^t_{\max} = \frac{1-\ell^t}{\nu^t(b^t)}\mathbb{I}_{\{b^t\}}$
3: **Output:** Learning sequence $(\mu^t, \nu^t)$.

---

policies are regularized proportionally to the learning rate. This gives the two policies $\mu^t$ and $\nu^t$ that are used to sample $\ell^t$. Then a regular mirror step update (with the Kullback-Leibler divergence) is applied with the unbiased estimate of the loss. The whole procedure is summarized in Algorithm 2.

These updates allow a relatively simple bound on the Kullback-Leibler divergence between the intermediate profiles $(\mu^{\tau,t}, \nu^{\tau,t})$ and the regularized Nash equilibrium, proven in Appendix C.

**Lemma 7.1.** *Let $\tau \in (0, 1]$. Taking $\eta^t = 2/(\tau(t+1))$ along with $\mu^{\tau,1} = \mu^0$ and $\nu^{\tau,1} = \nu^0$ the uniform policies gives with the above updates*

$$\mathbb{E}\left[\mathcal{D}_{KL}(\mu^{\tau,\star}, \mu^{\tau,t}) + \mathcal{D}_{KL}(\nu^{\tau,\star}, \nu^{\tau,t})\right] \leq \frac{2(A+B)}{\tau^2 t}$$

*for all $t \in \mathbb{N}$.*

To our knowledge, this is the first result of a $\mathcal{O}(1/(\tau^2 T))$ rate for the Kullback-Leibler divergence between the regularized solution of a game and the iterates under the bandit setting for a matrix game. This improves the rate $\mathcal{O}(1/\tau\sqrt{t})$ obtained by Dong et al. (2024), although this latter result was obtained with high probability (and not only in expectation), in addition of holding for *any* $\tau$-strongly monotone operator.

Using Pinsker inequality (Pinsker, 1964), this results in a $\mathcal{O}(1/(\tau\sqrt{T}))$ bound for the 1-norm between $(\mu^{\tau,t}, \nu^{\tau,t})$ and $(\mu^{\tau,\star}, \nu^{\tau,\star})$. Considering that the gap between $L^\tau$, for which $(\mu^{\tau,\star}, \nu^{\tau,\star})$ is optimal, and that $L$ is at most proportional to $\tau$, the best value of $\tau$ for the trade-off seems to be obtained when $\tau \asymp 1/(\tau\sqrt{T})$ where $\asymp$ denotes asymptotic equivalence up to a constant. This corresponds to $\tau \asymp T^{-1/4}$,

The following theorem, proven in Appendix C, formalizes this idea and provides guarantees for the final output of Algorithm 2.

**Theorem 7.2.** *Let $T$ be a fixed horizon. Then, using regularization $\tau = ((A + B)/T)^{1/4} \sqrt{2/(\log(A) + \log(B))}$ and the learning rates $\eta^t = 2/(\tau(t+1))$ for all $t$, Algorithm 2 guarantees*

$$\|EG(\mu^T, \nu^T)\|_2 \leq 3\sqrt{2}\left(\frac{A+B}{T}\right)^{1/4}\sqrt{\log(AB)}.$$

With the correct choice of regularization, Algorithm 2 therefore benefits from an optimal $L^2$ rate of $\mathcal{O}(T^{-1/4})$ for the final output. However, as explained above, this choice of regularization depends on the horizon $T$, and the above rate is thus only guaranteed near the final output. The actual last-iterate guarantee of the whole sequence up to $T$ is given by

$$f^t \asymp \min\left(1, \frac{T^{1/4}}{t^{1/2}}\right)$$

which technically does not match the lower bound of Theorem 5.1.

*Remark* 7.3. An intuitive way of fixing this issue would be to use an adaptive regularization parameter $\tau^t \asymp t^{-1/4}$ that decreases over time. Unfortunately, we failed to show the convergence of this method. The main reason is that, while the regularized solution can be shown to converge to the minimax profile minimizing the entropy as $\tau^t$ goes to 0, the convergence is sometimes too slow when one of the two minimax policies is on the border of the simplex. This can be interpreted through the perspectives of Azizian et al. (2021): the Legendre exponent of the entropy is different on the border.

## 8. Doubling trick and regularization

In order to get the anytime guarantees required for Algorithm 2, the usual solution in the online learning literature is the doubling trick. It consists in starting the algorithm with a small horizon $T_1$, and recursively restarting it every time the horizon is reached with a new horizon $T_i = 2T_{i-1}$. However, its use is not straightforward for this problem, as a complete restart of the algorithm would go directly against the last-iterate convergence assumption.

For this reason, we propose to use the doubling trick with a slight adjustment. Instead of directly restarting with a weaker regularization and discarding the current iterate after every subloop $i$, the meta-procedure will perform a trade-off between playing the old instance $i-1$, which has already been played over many iterations, and the new instance $i$, which has better asymptotical guarantees. Specifically, it will play the new instance with a certain probability $p_i^j$, close to 0 at the beginning and increasing to 1 over the iteration

$j$ of subloop $i$ as the new instance gets played more and obtains better guarantees.

This meta-procedure is summarized in Algorithm 3. As in Algorithm 1, the two players need to be synchronized in their choice between the old and the new instances, hence the same seed is sampled for both at the initialization.

The following theorem gives the rate of this meta-procedure.

**Theorem 8.1.** *Using Algorithm 3 with, for each $\mathcal{A}_i$, the algorithms and parameterization of Theorem 7.2, along with $T_i = 32(i2^i)$ and $S_i = 8(2^i)$, we obtain the bound*

$$\|EG(\mu^t, \nu^t)\|_2 \le 30 \left( \frac{A+B}{t} \right)^{1/4} \sqrt{\log(AB)\log(t)}$$

*for any total number of iterations $t = T_1 + ... + T_{i-1} + j$.*

The proof is given in Appendix C, and relies on a careful choice of the probabilities $p_i^j$ at each loop $i$ to compensate for the imperfect anytime guarantees of Algorithm 2. More precisely, this probability $p_i^j$ must be inversely proportional to the squared exploitability gap of the new iterates. This squared exploitability gap is roughly given by the Kullback-Leibler divergence between the iterates and the new regularized solution, whose inverse is proportional to the number of iterations from Lemma 7.1. This justifies an exponential increase up to 1, after which the procedure can be safely restarted.

**Synchronisation** Algorithm 3 therefore reaches the optimal rate $\widetilde{\mathcal{O}}(T^{-1/4})$ for the $L^2$ last-iterate convergence, up to logarithmic and constant factors. However, it shares some of the weaknesses of Algorithm 1, as it still requires the two players to synchronize their choices of either instance $\mathcal{A}_i$ or $\mathcal{A}_{i-1}$, with the latter being only played for the anytime guarantees.

## 9. Conclusion

We studied the convergence of uncoupled algorithms for learning zero-sum games with bandit feedback. We showed that imposing anytime last-iterate convergence worsens the rate compared to just requiring convergence of the average policies, with a lower bound of $\Omega(T^{-1/(2+p)})$ for the $L^p$ $F$-norm of the exploitability gap given $p \in (0, 2]$, in contrast of the usual rate $\mathcal{O}(T^{-1/2})$.

We then proposed two algorithms that match this rate. The first relies on some synchronization of the two players to balance between efficient exploration of the game and exploitation of a near-optimal policy. A second algorithm relies instead on a regularization of the game and also matches the rate for the $L^2$ norm. However, the latter does not have the required anytime guarantees. A doubling trick effectively solves this issue, using a synchronization similar to the first algorithm.

---

**Algorithm 3** Doubling trick approach

1: **Input:**
   Algorithms $\mathcal{A}_0, \mathcal{A}_1, \mathcal{A}_2, ...$
   Horizons $T_1, T_2, ... \in \mathbb{N}$
   Parameters $S_1, S_2, ... \in \mathbb{R}_{>0}$
2: **Initialize: Draw** $u \sim \mathcal{U}([0,1])$ common to both players
3: **Algorithm: For** $i = 1$ to $+\infty$:
   **For** $j = 1$ to $T_i$:
      $p_i^j \leftarrow \min\{1, \frac{1}{T_i} e^{\frac{j}{S_i}}\}$
      $k_i^j \leftarrow \lfloor \sum_{l=1}^j p_i^l + u \rfloor$
      **If** $k_i^j > k_i^{j-1}$:
         **Play** one iteration of $\mathcal{A}_i$
      **Otherwise**:
         **Play** one iteration of $\mathcal{A}_{i-1}$
4: **Output:** Learning sequence of policies with $L^2$ last-iterate convergence

---

This article opens the following research directions:

**Extensive-form games:** These results could be extended to the more general setting of extensive-form games (Kuhn, 1953), in which the players take multiple successive actions without complete knowledge of the current game state. The two approaches proposed in this article could be adapted in a relatively straight-forward way (using IXOMD (Kozuno et al., 2021) as algorithm $\mathcal{A}$ for the first, and the dilated Shannon entropy (Kroer et al., 2015) as the regularizer for the second) and obtain the same rate with respect to the horizon $T$. However, an interesting question arises: what is the optimal dependence on the total size of the action sets? This becomes particularly important in the context of extensive-form games where the number of actions [2] is very large.

**More natural methods:** Is it possible to obtain anytime last-iterate guarantees without relying on some synchronicity between the two players?

**Stronger convergence:** Is the $\Omega(T^{-1/4})$ lower bound tight for the uncoupled $L^p$ convergence given $p > 2$, and for the uncoupled convergence with high probability?

We especially wonder if these last two questions could be solved using a modified version of the optimistic mirror descent algorithm, for example by tweaking the estimated losses or the learning rates.

## Impact statement

This paper presents work whose goal is to advance the field of Machine Learning. There are many potential societal consequences of our work, none which we feel must be

---

[2]The number of state-action pairs to be precise.

specifically highlighted here.

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

## A. Lower bound

**Theorem 5.1.** *Assume that the sampled action $b^t \sim \nu^t$ of the max-player at each iteration $t$ is never observed. Then, for any $p \in (0, 2]$, no learning sequence can achieve simultaneously for all $2 \times 2$ matrix games described above an $L^p$ last iterate convergence of*

$$f^t = \frac{4^{\frac{-1}{p}}}{118} t^{\frac{-1}{2+p}},$$

*even asymptotically.*

*Proof.* We will use the games mentioned in the main body, defined with:

$$M^\varepsilon = \begin{bmatrix} 2/3 - \varepsilon & 1/3 + \varepsilon \\ 1/3 & 2/3 \end{bmatrix} \qquad \text{with } \varepsilon \in [-1/12, 1/12].$$

where the entries of the matrix indicate the parameters of some i.i.d Bernoulli for the actual game.

We will assume that the learning sequence satisfies an asymptotic $L^p$ last iterate convergence with $f$ of the form

$$f^t = C_p \, t^{\frac{-1}{2+p}} \quad \text{with } C_p \in (0, 1],$$

and show that this is impossible for all games given $C_p$ low enough.

**Notations**   Let $z^t$ denote the knowledge of both players up to round $t$ included with $z^t := \left( \omega, a^1, \ell^1, ..., a^t, \ell^t \right)$, such that the sequences $(\mu^t)$ and $(\nu^t)$ of policies are predictable with respect to the filtration $(\sigma(z^t))_{t \in \mathbb{N}}$ rather than $\mathcal{F}$ from the assumption.

We will consider the games $M^0$, $M^{\varepsilon_T}$ and $M^{-\varepsilon_T}$ with a fixed choice of $\varepsilon_T$ that we will specify later. Let $\mathbb{P}_0$, $\mathbb{P}_T$ and $\mathbb{P}_{-T}$ respectively be the probabilities associated to playing these games, with $\|\cdot\|_{0, p}$, $\|\cdot\|_{T, p}$ and $\|\cdot\|_{-T, p}$ the associated $p$ (pseudo)-norms. For any random variable $X$ and $\theta \in \mathbb{Z}$, we will denote by $\mathbb{P}_\theta^X$ the probability distribution on $X$ given $\mathbb{P}_\theta$.

Now, given an horizon $T$, let $E^T$ and $E^{-T}$ respectively be the two events $E^T := \left\{ \mu^T(1) \leq 1/2 \right\}$ and $E^{-T} := \left\{ \mu^T(1) \geq 1/2 \right\}$. $E^T$ denotes a suboptimal choice for the game $M^{\varepsilon_T}$, and $E^{-T}$ a suboptimal choice for the game $M^{-\varepsilon_T}$ as we will show in the next section. Under $P_0$, one of these two events happens with a probability of at least $1/2$, we will denote it with $\theta^T \in \{-T, T\}$

For clarity, we will take the strategy of the max-player $\nu^t$ to be in the form $\begin{bmatrix} 1/2 + \delta^t \\ 1/2 - \delta^t \end{bmatrix}$ with $\delta^t \in [-1/2, 1/2]$..

**Link between the policies and exploitability gaps**   The exploitability gap will be lower bounded twice in this section: for the max-player under the game $M^0$ and for the min-player under the games $M^{\varepsilon^T}$ and $M^{-\varepsilon^T}$. We will use the fact that the value of all of these games is $1/2$.

Given a max-player policy $\nu^t$, we have

$$M^0 . \nu^t = \begin{bmatrix} 1/2 + \delta^t/3 \\ 1/2 - \delta^t/3 \end{bmatrix}$$

which implies (for $M^0$), $EG(\mu^t, \nu^t) \geq 1/2 - \min_\mu \left\langle \mu, M^0 . \nu^t \right\rangle = \delta^t/3$

On the other hand, given a min-player policy $\mu^T = \begin{bmatrix} \mu^T(1) \\ 1 - \mu^T(1) \end{bmatrix}$ and any $\varepsilon \in [-1/12, 1/12]$,

$$(M^\varepsilon)^T . \mu^T = \begin{bmatrix} 1/3 + 1/3\mu^T(1) - \varepsilon\mu^T(1) \\ 2/3 - 1/3\mu^T(1) + \varepsilon\mu^T(1) \end{bmatrix}.$$

(with the $M^T$ notation for the transpose). Then, using the vectors $e^1 = \begin{bmatrix} 1 \\ 0 \end{bmatrix}$ and $e^2 = \begin{bmatrix} 0 \\ 1 \end{bmatrix}$,

With $M^{\varepsilon_T}$ under event $E^T = \{\mu^T(1) \leq 1/2\}:$ $\quad \left\langle e^2, (M^{\varepsilon_T})^T . \mu^T \right\rangle \geq 1/2 + \varepsilon_T/2,$

hence: $\quad EG(\mu^t, \nu^t) \geq \max_\nu \left\langle \nu, (M^{\varepsilon_T})^T . \mu^T \right\rangle - 1/2 \geq \varepsilon_T/2$

With $M^{-\varepsilon_T}$ under event $E^{-T} = \{\mu^T(1) \geq 1/2\}:$ $\quad \left\langle e^1, (M^{-\varepsilon_T})^T . \mu^T \right\rangle \geq 1/2 + \varepsilon_T/2,$

hence: $\quad EG(\mu^t, \nu^t) \geq \max_\nu \left\langle \nu, (M^{-\varepsilon_T})^T . \mu^T \right\rangle - 1/2 \geq \varepsilon_T/2$

In both cases, the exploitability gap is at least $\varepsilon^t/2$.

**Asymptotic KL divergence** From the definition of the sequence $(z^t)$ and the additive properties of the KL divergence, we obtain the following inequalities

$$
\begin{aligned}
\mathcal{D}_{\mathrm{KL}}\left(\mathbb{P}_0^{z^T}, \mathbb{P}_{\theta^T}^{z^T}\right) &= \mathcal{D}_{\mathrm{KL}}\left(\mathbb{P}_0^{z^0}, \mathbb{P}_{\theta^T}^{z^0}\right) + \mathbb{E}_0\left[\sum_{t=1}^T \mathcal{D}_{\mathrm{KL}}\left(\mathbb{P}_0^{z^t|z^{t-1}}, \mathbb{P}_{\theta^T}^{z^t|z^{t-1}}\right)\right] \\
&= \mathcal{D}_{\mathrm{KL}}\left(\mathbb{P}_0^\omega, \mathbb{P}_{\theta^T}^\omega\right) + \mathbb{E}_0\left[\sum_{t=1}^T \mathcal{D}_{\mathrm{KL}}\left(\mathbb{P}_0^{a^t|z^{t-1}}, \mathbb{P}_{\theta^T}^{a^t|z^{t-1}}\right)\right] + \mathbb{E}_0\left[\sum_{t=1}^T \mathcal{D}_{\mathrm{KL}}\left(\mathbb{P}_0^{\ell^t|z^{t-1}}, \mathbb{P}_{\theta^T}^{\ell^t|z^{t-1}}\right)\right] \\
&= \mathbb{E}_0\left[\sum_{t=1}^T \mu^t(1)\mathcal{D}_{\mathrm{KL}}\left(1/2 + \delta^t/3, 1/2 + \delta^t/3 - 2\delta^t\mathrm{sign}(\theta^T)\varepsilon_T\right)\right] \\
&\leq 24\varepsilon_T^2\mathbb{E}_0\left[\sum_{t=1}^T |\delta^t|^2\right] \\
&\leq 24\varepsilon_T^2\mathbb{E}_0\left[\sum_{t=1}^T |\delta^t|^p\right] \\
&\leq 24\varepsilon_T^2\mathbb{E}_0\left[\sum_{t=1}^T \left(3EG(\mu^t, \nu^t)\right)^p\right] \\
&\leq 216\varepsilon_T^2 \sum_{t=1}^T \|EG(\mu^t, \nu^t)\|_{0,p}^p
\end{aligned}
$$

where we especially used the reversed Pinsker inequality with $1/6 \leq 1/2 + \delta^t/3 - 2\delta^t\mathrm{sign}(\theta^T)\varepsilon_T \leq 5/6$, and the previous lower bound of the exploitability gap under $\mathbb{P}_0$. For the third equality:

- $\mathcal{D}_{\mathrm{KL}}\left(\mathbb{P}_0^\omega, \mathbb{P}_{\theta^T}^\omega\right)$ is 0 as the internal randomness do not depend on the model.

- $\mathcal{D}_{\mathrm{KL}}\left(\mathbb{P}_0^{a^t|z^{t-1}}, \mathbb{P}_{\theta^T}^{a^t|z^{t-1}}\right)$ is also 0, for all $t$, as the choice of the action (conditionally on the previous move) also does not depend on the model.

- From the computations above, $\mathcal{D}_{\mathrm{KL}}\left(\mathbb{P}_0^{\ell^t|z^{t-1}}, \mathbb{P}_{\theta^T}^{\ell^t|z^{t-1}}\right)$ corresponds to the KL divergence between a Bernoulli $\mathcal{B}(1/2 + \delta^t/3)$ and a Bernoulli $\mathcal{B}(1/2 + \delta^t/3 - 2\delta^t\mathrm{sign}(\theta^T)\varepsilon_T)$ if the first move is played by the first player, 0 otherwise.

Now assume that the restriction holds for all $t > T_0$, then for any $T > T_0$

$$\mathcal{D}_{\mathrm{KL}}\left(\mathbb{P}_0^{z^T}, \mathbb{P}_{\theta^T}^{z^T}\right) \leq 216\varepsilon_T^2 \left(T_0 + \sum_{t=T_0+1}^{T} (f^t)^p\right)$$

$$\leq 216\varepsilon_T^2 \left(T_0 + \sum_{t=1}^{T} t^{\frac{-p}{2+p}}\right)$$

$$\leq 216\varepsilon_T^2 \left(T_0 + \int_0^T t^{\frac{2}{2+p}-1} dt\right)$$

$$= 216\varepsilon_T^2 \left(T_0 + \frac{2+p}{2} T^{\frac{2}{2+p}}\right)$$

$$\leq 216\varepsilon_T^2 \left(T_0 + 2\, T^{\frac{2}{2+p}}\right).$$

Which implies the asymptotic bound:

$$\limsup_{T \to +\infty} \mathcal{D}_{\mathrm{KL}}\left(\mathbb{P}_0^{z^T}, \mathbb{P}_{\theta^T}^{z^T}\right) T^{\frac{-2}{2+p}} \leq 432\varepsilon_T^2.$$

**Probability of a suboptimal choice**   Defining $\|\cdot\|_{TV}$ the total variation between two probabilities $\mathbb{P}$ and $\mathbb{Q}$:

$$\|\mathbb{P} - \mathbb{Q}\|_{TV} = \sup_{E \text{ event}} |\mathbb{P}(E) - \mathbb{Q}(E)|$$

and using the measurability of $E^{\theta^T}$ with respect to $\sigma\left(z^T\right)$, the fact that $\mathbb{P}_0(E^{\theta^T}) \geq 1/2$ by definition and Pinsker inequality, we get

$$\liminf_{T \to +\infty} \mathbb{P}_{\theta^T}(E^{\theta^T}) \geq \liminf_{T \to +\infty} \left[\mathbb{P}_0(E^{\theta^T}) - \left\|\mathbb{P}_0^{z^T} - \mathbb{P}_{\theta^T}^{z^T}\right\|_{TV}\right]$$

$$\geq \frac{1}{2} - \limsup_{T \to +\infty} \sqrt{\frac{1}{2}\mathcal{D}_{\mathrm{KL}}\left(\mathbb{P}_0^{z^T}, \mathbb{P}_{\theta^T}^{z^T}\right)}$$

$$\geq \frac{1}{2} - 6\sqrt{6} \limsup_{T \to +\infty} \varepsilon_T T^{\frac{1}{2+p}}$$

.

This implies that fixing $\varepsilon_T = \frac{1}{24\sqrt{6}} T^{\frac{-1}{2+p}}$ forces the probability of this suboptimal choice to be at least $\frac{1}{4}$ asymptotically.

**Final bound**   Combining the previous choice of $\varepsilon^T$ with the suboptimality of $\mu^T$ under $E^{\theta^T}$,

$$\liminf_{T \to +\infty} \|EG(\mu^T, \nu^T)\|_{\theta^T, p} T^{\frac{1}{2+p}} \geq \liminf_{T \to +\infty} \left(\mathbb{P}_{\theta^T}\left(E^{\theta^T}\right) \mathbb{E}_{\theta^T}\left[\left(EG(\mu^T, \nu^T)^p \Big| E^{\theta^T}\right)\right]\right)^{\frac{1}{p}} T^{\frac{1}{2+p}}$$

$$\geq \liminf_{T \to +\infty} \mathbb{P}_{\theta^T}\left(E^{\theta^T}\right)^{\frac{1}{p}} \frac{\varepsilon_T}{2} T^{\frac{1}{2+p}}$$

$$= \frac{1}{48\sqrt{6}} \liminf_{T \to +\infty} \mathbb{P}_{\theta^T}\left(E^{\theta^T}\right)^{\frac{1}{p}}$$

$$\geq \frac{4^{\frac{-1}{p}}}{48\sqrt{6}}.$$

As $\frac{1}{48\sqrt{6}} > \frac{1}{118}$, this implies that the guarantees cannot be attained asymptotically for all games assuming $f$ is defined with

$$f^t = \frac{4^{\frac{-1}{p}}}{118} t^{\frac{-1}{2+p}}.$$

$\square$

## B. Proofs for the Simultaneous Explore or Exploit approach

**Lemma B.1.** *Let $(p^t) \in [0,1]^{\mathbb{N}}$. The sequence of Bernoulli random variables $(B^t)_{t \in \mathbb{N}}$ defined by*

$$B^t = \lfloor s^t + u \rfloor - \lfloor s^{t-1} + u \rfloor \quad where \quad s^t = \sum_{i=1}^{t} p^i \quad and \quad u \sim \mathcal{U}([0,1])$$

*satisfies:*

$$\forall t \in \mathbb{N}, \quad \mathbb{E}[B^t] = p^t \quad and \quad \sum_{i=1}^{t} B^i \geq \lfloor s^t \rfloor .$$

*Proof.* Let $x, y \in \mathbb{R}$ such that $0 \leq y - x \leq 1$. Let $\{\cdot\}$ denote the fractional part. We distinguish two cases with the same result:

- If $\{x\} \leq \{y\}$ (and consequently $\lfloor x \rfloor = \lfloor y \rfloor$), then

$$\mathbb{P}\left(\lfloor x + u \rfloor + 1 = \lfloor y + u \rfloor\right) = \mathbb{P}\left(u \in [1 - \{y\}, 1 - \{x\})\right) = \{y\} - \{x\} = y - x$$

- If $\{x\} > \{y\}$ (which necessarily implies $\lfloor x \rfloor = \lfloor y \rfloor - 1$),

$$\mathbb{P}\left(\lfloor x + u \rfloor + 1 = \lfloor y + u \rfloor\right) = \mathbb{P}\left(u \in [0, 1 - \{x\}) \cup [1 - \{y\}, 1]\right) = 1 - \{x\} + \{y\} = y - x .$$

Using, for each $t \in \mathbb{N}$, $x = s^{t-1}$ and $y = s^t$ yields the first equality. The inequality is obtained by telescoping the term:

$$\sum_{i=1}^{T} B^i = \lfloor s^T + u \rfloor - \lfloor u \rfloor \geq \lfloor s^T \rfloor$$

. $\square$

**Lemma B.2.** *Assume that $\mathcal{A}$ satisfies an output convergence of rate $g$. Then Algorithm 1 satisfies an $L^p$ last iterate convergence of rate $f$ defined by*

$$f^t = 2^{\frac{1}{p}} \left[ (p^t)^{\frac{1}{p}} + g^{r^t} \right]$$

*where $r^t = \left\lfloor \sum_{k=1}^{t} p^k \right\rfloor$.*

*Proof.* Using the previous lemma where $B^t$ is the action of exploring, we obtain as $g$ is non-increasing,

$$\begin{aligned}
\|EG(\mu^t, \nu^t)\|_p &= \mathbb{E}\left[EG(\mu^t, \nu^t)^p\right]^{\frac{1}{p}} \\
&= \left(p^t \mathbb{E}\left[EG(\mu^t, \nu^t)\right] + (1 - p^t)\mathbb{E}\left[EG(\hat{\mu}^{k^t}, \hat{\nu}^{k^t})\right]\right)^{\frac{1}{p}} \\
&\leq \left(p^t + \left(g^{r^t}\right)^p\right)^{\frac{1}{p}} \\
&\leq 2^{\frac{1}{p}} \left[(p^t)^p + g^{r^t}\right]
\end{aligned}$$

where we used the notation $k^t = \lfloor \sum_{k=1}^{t} p^k + u \rfloor$ and the inequalities, for all $x, y \in \mathbb{R}_{>0}$:

$$(x + y)^q \leq 2^{q-1}(x^q + y^q) \quad \text{for } q \geq 1 \quad \text{from the convexity of } x^q$$
$$(x + y)^q \leq (x^q + y^q) \quad \text{for } q < 1 \quad \text{as } y \mapsto (x + y)^q - x^q - y^q \text{ is decreasing on } \mathbb{R}_{>0}.$$

$\square$

**Theorem B.3.** *(Neu, 2015) Given $A$ arms and $\delta \in (0,1)$, setting $\eta^t = 2\gamma^t = \sqrt{\frac{\log A}{At}}$ for all $t$, the bound of EXP3-IX is*

$$R^T \le 4\sqrt{AT \log(A)} + \left(2\sqrt{\frac{AT}{\log A}} + 1\right) \log(2/\delta)$$

*with probability $1 - \delta$*

Applying this theorem to both $R^t_{\min}$ and $R^t_{\max}$ with $\delta' = \delta/2$ gives the bound, with probability at least $1 - \delta$:

$$R^t_{\min} + R^t_{\max} \le 4\sqrt{At \log(A)} + \left(2\sqrt{\frac{At}{\log A}} + 1\right) \log(4/\delta) + 4\sqrt{Bt \log(B)} + \left(2\sqrt{\frac{Bt}{\log B}} + 1\right) \log(4/\delta)$$

$$\le 8\sqrt{(A+B)t} \log(A+B) + \left(2\sqrt{At} + 2\sqrt{Bt}\right) \log(4/\delta)$$

$$\le 8\sqrt{(A+B)t} \log(4(A+B)/\delta)$$

where we used very loose upper bounds for simplicity.

**Theorem 6.2.** *Using Algorithm 1 with probabilities $p^t = t^{-p/(2+p)}$, and as $\mathcal{A}$, the algorithm EXP3-IX with parameters $\eta^t_{min} = 2\gamma^t_{min} = \sqrt{\log(A)/(At)}$ for the min-player and $\eta^t_{max} = 2\gamma^t_{max} = \sqrt{\log(B)/(Bt)}$ for the max-player, we obtain an $L^p$ last-iterate convergence rate of*

$$f^t = 17\sqrt{A+B} \, 2^{\frac{1}{p}} t^{-\frac{1}{2+p}} \log\left(4(A+B)t^2/p\right).$$

*Proof.* **$L^p$ bound**

We first show that the average policy played by EXP3-IX converges for the $L^p$ norm. Let $p \in (0, 2]$ and $t \in \mathbb{N}$, we consider $\delta = p/t^p$ in the above inequality. Then, as the sum of the two regrets is bounded by $T$,

$$\|EG(\hat{\mu}^t, \hat{\nu}^t)\|_p = \mathbb{E}\left[EG(\hat{\mu}^t, \hat{\nu}^t)^p\right]^{\frac{1}{p}}$$

$$\le \frac{1}{t}\left[\left(R^t_{\min} + R^t_{\max}\right)^p\right]^{\frac{1}{p}}$$

$$\le \frac{1}{t}\left[\delta t^p + (1-\delta)\left(8\sqrt{(A+B)t} \log(4(A+B)/\delta)\right)^p\right]^{\frac{1}{p}}$$

$$\le \frac{1}{t}\left[p + \left(8\sqrt{(A+B)t} \log(4(A+B)t^2/p)\right)^p\right]^{\frac{1}{p}}.$$

As for all $x \ge 1$, from the concavity of the log function,

$$(p + x^p)^{\frac{1}{p}} = e^{\frac{1}{p}\log(p+x^p)} \le e^{\frac{1}{p}\log(x^p)+1} = ex,$$

we obtain

$$\|EG(\hat{\mu}^t, \hat{\nu}^t)\|_p \le 6\sqrt{\frac{A+B}{t}} \log\left(4(A+B)t^2/p\right)$$

**Lemma application** We apply Lemma 6.1 with $p^t = t^{-\frac{p}{2+p}}$. As in this case,

$$\lfloor r^t \rfloor \ge \left\lfloor \int_1^{t+1} u^{-\frac{p}{2+p}} du \right\rfloor \ge \left\lfloor \frac{2+p}{p}\left((t+1)^{\frac{2}{2+p}} - 1\right) \right\rfloor \ge 2\left(t^{\frac{2}{2+p}} - 2\right).$$

We obtain (loosely) a rate

$$f^t = 2^{\frac{1}{p}}\left[(p^t)^{\frac{1}{p}} + g^{r^t}\right]$$

$$\le 2^{\frac{1}{p}}\left[t^{-\frac{1}{2+p}} + 11\sqrt{2}\sqrt{A+B} \log\left(4(A+B)t^2/p\right) t^{-\frac{1}{2+p}}\right]$$

$$\le 2^{\frac{1}{p}} 17\sqrt{A+B} \log\left(4(A+B)t^2/p\right) t^{-\frac{1}{2+p}}.$$

$\square$

## C. Proofs for the Regularized EXP3 algorithm

**Notations:** In this section, the computations will be done directly in the space $W = \Delta_A \times \Delta_B$ using $w = (\mu, \nu)$ and the operators, for $\tau > 0$,

$$F(w) = (L(\cdot, \nu), 1 - L(\mu, \cdot))$$

$$\hat{F}^t = (\hat{\ell}_{\min}^t, \hat{\ell}_{\max}^t)$$

$$h(w) = \sum_{a=1}^A \mu(a) \left(\log(\mu(a)) - 1\right) + \sum_{b=1}^B \nu(b) \left(\log(\nu(b)) - 1\right)$$

$$F^\tau(w) = (L(\cdot, \nu), 1 - L(\mu, \cdot)) + \tau \left(\nabla h(w) - \nabla h(w^0)\right)$$

$$\mathcal{D}(w, w') = \mathcal{D}_{\mathrm{KL}}(\mu, \mu') + \mathcal{D}_{\mathrm{KL}}(\nu, \nu').$$

With these notations, we have the following properties:

- $F$ is a monotone operator over $W$, and the solutions $w^\star \in W$ of the variational inequalities

$$\forall w \in W, \langle F(w^\star), w^\star - w \rangle \leq 0$$

  are the Nash equilibrium of the game associated to $L$.

- $F^\tau$ is a strongly monotone operator. It has only one solution $w^{\tau,\star}$, for which the above inequality is always an equality and satisfies for all $w, w' \in W$:

$$\langle F^\tau(w) - F^\tau(w'), w - w' \rangle = \tau \left(D(w, w') + D(w', w)\right)$$

- $\mathbb{E}(\hat{F}^t | \mathcal{F}^{t-1}) = F(w^t)$

- $\mathcal{D}$ is the Bregman divergence associated to $h$, and especially satisfies the law of cosines, for all $x, y, z \in W$:

$$\mathcal{D}(x, z) = \mathcal{D}(x, y) + \mathcal{D}(y, z) + \langle \nabla h(z) - \nabla h(y), y - x \rangle$$

Before proving the lemma, we can notice that the updates of Algorithm 2 can be rewritten in the $W$ space (if $\tau \eta^t \leq 1$):

$$w^t = \arg\min_{w \in W} (1 - \eta^t \tau) D(w, w^{\tau,t}) + \eta^t \tau D(w, w^0)$$

$$w^{\tau,t+1} = \arg\min_{w \in W} \eta^t \left\langle \hat{F}^t, w \right\rangle + D(w, w^t)$$

which are equivalent to the updates, taking the gradient of the above expressions,

$$\nabla h(w^t) \equiv (1 - \tau \eta^t) \nabla h(w^{\tau,t}) + \tau \eta^t \nabla h(w^0)$$

$$\nabla h(w^{\tau,t+1}) \equiv \nabla h(w^t) - \eta^t \hat{F}^t$$

where we used the notations

$$x \equiv y \iff \forall w \in W, \langle x - y, w \rangle = 0$$

**Lemma C.1.** *Let $\tau \in (0, 1]$. Taking $\eta^t = 2/(\tau(t+1))$ along with $\mu^{\tau,1} = \mu^0$ and $\nu^{\tau,1} = \nu^0$ the uniform policies gives with the above updates*

$$\mathbb{E}\left[\mathcal{D}_{KL}(\mu^{\tau,\star}, \mu^{\tau,t}) + \mathcal{D}_{KL}(\nu^{\tau,\star}, \nu^{\tau,t})\right] \leq \frac{2(A + B)}{\tau^2 t}$$

*for all $t \in \mathbb{N}$.*

*Proof.* **Updates:** From the law of cosines applied to the two updates, we have:

$$
\begin{aligned}
(1) \quad (1 - \eta^t \tau)\mathcal{D}(w^{\tau,\star}, w^{\tau,t}) &= (1 - \eta^t \tau)\mathcal{D}(w^{\tau,\star}, w^t) + (1 - \eta^t \tau)\mathcal{D}(w^t, w^{\tau,t}) \\
&\quad + (1 - \eta^t \tau)\left\langle \nabla h(w^{\tau,t}) - \nabla h(w^t), w^t - w^{\tau,\star} \right\rangle \\
&\geq (1 - \eta^t \tau)\mathcal{D}(w^{\tau,\star}, w^t) + \eta^t \tau \left\langle \nabla h(w^t) - \nabla h(w^0), w^t - w^{\tau,\star} \right\rangle
\end{aligned}
$$

and

$$
\begin{aligned}
(2) \quad \mathcal{D}(w^{\tau,\star}, w^{\tau,t+1}) &= \mathcal{D}(w^{\tau,\star}, w^t) + \mathcal{D}(w^t, w^{\tau,t+1}) + \left\langle \nabla h(w^{\tau,t+1}) - \nabla h(w^t), w^t - w^{\tau,\star} \right\rangle \\
&= \mathcal{D}(w^{\tau,\star}, w^t) + \mathcal{D}(w^t, w^{\tau,t+1}) - \eta^t \left\langle \hat{F}^t, w^t - w^{\tau,\star} \right\rangle.
\end{aligned}
$$

$(2) - (1)$ yields when conditioned on $\mathcal{F}^{t-1}$:

$$
\begin{aligned}
\mathbb{E}\left[\mathcal{D}(w^{\tau,\star}, w^{\tau,t+1}) | \mathcal{F}^{t-1}\right] &\leq (1 - \eta^t \tau)\mathcal{D}(w^{\tau,\star}, w^{\tau,t}) + \mathbb{E}\left[\mathcal{D}(w^t, w^{\tau,t+1}) | \mathcal{F}^{t-1}\right] \\
&\quad + \eta^t \tau \mathcal{D}(w^{\tau,\star}, w^t) - \eta^t \left\langle w^t - w^{\tau,\star}, F^\tau(w^t) \right\rangle \\
&= (1 - \eta^t \tau)\mathcal{D}(w^{\tau,\star}, w^{\tau,t}) + \mathbb{E}\left[\mathcal{D}(w^t, w^{\tau,t+1}) | \mathcal{F}^{t-1}\right] \\
&\quad + \eta^t \tau \mathcal{D}(w^{\tau,\star}, w^t) - \eta^t \left\langle w^t - w^{\tau,\star}, F^\tau(w^t) - F^\tau(w^{\tau,\star}) \right\rangle \\
&= (1 - \eta^t \tau)\mathcal{D}(w^{\tau,\star}, w^{\tau,t}) + \mathbb{E}\left[\mathcal{D}(w^t, w^{\tau,t+1}) | \mathcal{F}^{t-1}\right] \\
&\quad - \eta^t \tau \mathcal{D}(w^t, w^{\tau,\star}) \\
&\leq (1 - \eta^t \tau)\mathcal{D}(w^{\tau,\star}, w^{\tau,t}) + \mathbb{E}\left[\mathcal{D}(w^t, w^{\tau,t+1}) | \mathcal{F}^{t-1}\right],
\end{aligned}
$$

**Second order bound:** We now want to show

$$
\mathbb{E}\left[\mathcal{D}(w^t, w^{\tau,t+1}) | \mathcal{F}^{t-1}\right] \leq \left(\eta^t\right)^2 \frac{A+B}{2}.
$$

We start by showing

$$
\mathbb{E}\left[\mathcal{D}_{\mathrm{KL}}(\mu^t, \mu^{\tau,t+1}) | \mathcal{F}^{t-1}\right] \leq \left(\eta^t\right)^2 \frac{A}{2}
$$

and the rest will follow by symmetry. This inequality is classic in the bandit literature, but we provide a quick proof below for completeness.

If we define $\tilde{\mu}^{\tau,t+1}$ as the unprojected update defined by:

$$
\nabla h_{\mathcal{A}}(\tilde{\mu}^{\tau,t+1}) = \nabla h_{\mathcal{A}}(\mu^t) - \eta^t \hat{\ell}^t_{\min}
$$

we know, because of the generalized Pythagorean theorem (Hiriart-Urruty & Lemaréchal, 2001), that

$$
\mathbb{E}\left[\mathcal{D}_{\mathrm{KL}}(\mu^t, \mu^{\tau,t+1}) | \mathcal{F}^{t-1}\right] \leq \mathbb{E}\left[\mathcal{D}_{\mathrm{KL}}(\mu^t, \tilde{\mu}^{\tau,t+1}) | \mathcal{F}^{t-1}\right].
$$

Then, using the convex conjugate $h^\star$ (Hiriart-Urruty & Lemaréchal, 2001), the following holds, using classical properties of the Bregman divergence:

$$
\begin{aligned}
\mathbb{E}\left[\mathcal{D}_{\mathrm{KL}}(\mu^t, \tilde{\mu}^{\tau,t+1}) | \mathcal{F}^{t-1}\right] &= \mathbb{E}\left[\mathcal{D}_{h^\star_{\mathcal{A}}}(\nabla h_{\mathcal{A}}(\tilde{\mu}^{\tau,t+1}), \nabla h_{\mathcal{A}}(\mu^t)) | \mathcal{F}^{t-1}\right] \\
&= \mathbb{E}\left[\mathcal{D}_{h^\star_{\mathcal{A}}}(\nabla h_{\mathcal{A}}(\mu^t) - \eta^t \hat{\ell}^t_{\min}, \nabla h_{\mathcal{A}}(\mu^t)) | \mathcal{F}^{t-1}\right] \\
&\leq \frac{\left(\eta^t\right)^2}{2} \mathbb{E}\left[\left\langle \nabla^2 h^\star_{\mathcal{A}}(\nabla h(\mu^t)).\hat{\ell}^t_{\min}, \hat{\ell}^t_{\min} \right\rangle | \mathcal{F}^{t-1}\right] \\
&= \frac{\left(\eta^t\right)^2}{2} \sum_{a=1}^A \mu^t(a) \frac{\ell^t(a)^2}{\mu^t(a)^2} \mu^t(a) \\
&\leq \left(\eta^t\right)^2 \frac{A}{2}
\end{aligned}
$$

where we used Taylor inequality along with the fact that the hessian

$$\nabla^2 h_{\mathcal{A}}^\star(\nabla h_{\mathcal{A}}(\mu)) = \nabla^2 h_{\mathcal{A}}(\mu)^{-1} = \mathrm{Diag}(\mu(a))_{a=1,\dots,A}$$

is increasing on all components with respect to $\mu$, which implies the negativity of the third-order term. Indeed, each component of the unprojected policy decreases along the update.

**Recursion** We then have the recursive property, taking the global expectation:

$$\mathbb{E}\left[\mathcal{D}(w^{\tau,\star}, w^{\tau,t+1})\right] \leq (1 - \eta^t \tau)\mathbb{E}\left[\mathcal{D}(w^{\tau,\star}, w^{\tau,t})\right] + (\eta^t)^2 \frac{A+B}{2}.$$

With $\eta^t = \frac{2}{\tau(t+1)}$, this is everything we need for the desired property

$$\mathbb{E}\left[\mathcal{D}(w^{\tau,\star}, w^{\tau,t})\right] \leq 2\frac{A+B}{\tau^2 t}.$$

Indeed, for $t = 1$, the property immediately follows from, as $w^{\tau,1}$ is the uniform profile,

$$D(w^{\tau,\star}, w^{\tau,1}) = h(w^{\tau,\star}) \leq \log(A) + \log(B) \leq \frac{A+B}{\tau^2}$$

and $\tau \leq 1$ by assumption.

Then, assuming the property holds for $t$, we notice that for $t + 1$

$$\begin{aligned}
\mathbb{E}\left[\mathcal{D}(w^{\tau,\star}, w^{t+1})\right] &\leq 2\left(1 - \frac{2}{t+1}\right)\frac{A+B}{\tau^2 . t} + 2\frac{A+B}{\tau^2(t+1)^2} \\
&= 2\frac{A+B}{\tau^2}\left(\frac{1}{t} - \frac{2}{t(t+1)} + \frac{1}{(t+1)^2}\right) \\
&\leq 2\frac{A+B}{\tau^2}\left(\frac{1}{t} - \frac{1}{t(t+1)}\right) \\
&= 2\frac{A+B}{\tau^2(t+1)}.
\end{aligned}$$

$\square$

**Lemma C.2.** *For a zero-sum game with rewards in $(0,1)$, the exploitability gap is $1$-Lipchitz with respect to the $1$-norm.*

*Proof.* Let $w = (\mu, \nu)$ and $w' = (\mu', \nu')$ be two profiles in $\Delta_A \times \Delta_B$, then

$$\begin{aligned}
|EG(w) - EG(w')| &= \left|\sup_{\mu^\dagger} L(\mu^\dagger, \nu) - \sup_{\mu^\dagger} L(\mu^\dagger, \nu') - \sup_{\nu^\dagger} L(\mu, \nu^\dagger) + \sup_{\nu^\dagger} L(\mu', \nu^\dagger)\right| \\
&\leq \left|\sup_{\mu^\dagger} L(\mu^\dagger, \nu) - \sup_{\mu^\dagger} L(\mu^\dagger, \nu')\right| + \left|\sup_{\nu^\dagger} L(\mu, \nu^\dagger) - \sup_{\nu^\dagger} L(\mu', \nu^\dagger)\right| \\
&\leq \|\nu - \nu'\|_1 + \|\mu - \mu'\|_1 \\
&\leq \|w - w'\|_1
\end{aligned}$$

following the fact that, for any $(\mu^\dagger, \nu^\dagger)$, $\nu \mapsto L(\mu^\dagger, \nu)$ and $\mu \mapsto L(\mu, \nu^\dagger)$ are both $1$-Lipschitz as the coefficients of the matrix are in $[0,1]$. $\square$

**Theorem 7.2.** *Let $T$ be a fixed horizon. Then, using regularization $\tau = ((A+B)/T)^{1/4}\sqrt{2/(\log(A) + \log(B))}$ and the learning rates $\eta^t = 2/(\tau(t+1))$ for all t, Algorithm 2 guarantees*

$$\|EG(\mu^T, \nu^T)\|_2 \leq 3\sqrt{2}\left(\frac{A+B}{T}\right)^{1/4}\sqrt{\log(AB)}.$$

*Proof.* We can safely assume $T \geq A + B \geq 4$ as the bound is immediate otherwise. We first notice, at iteration $T$:

- From the optimality of $w^{\tau,\star}$ up to the regularization,

$$EG(w^{\tau,\star}) \leq \tau \mathcal{D}_{\mathrm{KL}}(w^{\tau,\star}, w^0) \leq \tau \left(\log(A) + \log(B)\right).$$

- From Pinsker inequality and Lemma 7.1,

$$\|w^{\tau,T} - w^{\tau,\star}\|_1^2 \leq 2\mathcal{D}_{\mathrm{KL}}(w^{\tau,\star}, w^{\tau,T}) \leq \frac{4(A+B)}{\tau^2 T}.$$

- By definition of $w^T$ as an $\arg\min$,

$$(1 - \eta^T\tau)\mathcal{D}_{\mathrm{KL}}(w^T, w^{\tau,T}) + \tau\eta^T \mathcal{D}_{\mathrm{KL}}(w^T, w^0) \leq \eta^T\tau\mathcal{D}_{\mathrm{KL}}(w^{\tau,T}, w^0).$$

Hence, re-using Pinsker inequality,

$$\|w^T - w^{\tau,T}\|_1^2 \leq 2\mathcal{D}_{\mathrm{KL}}(w^T, w^{\tau,T}) \leq 2\frac{\eta^T\tau}{1 - \eta^T\tau}\mathcal{D}_{\mathrm{KL}}(w^{\tau,t}, w^0) \leq 4\eta^T\tau\left(\log(A) + \log(B)\right) = \frac{8}{T+1}\left(\log(A) + \log(B)\right)$$

where we used $\eta^T = \frac{2}{\tau(T+1)}$, and in particular $\eta^T\tau \leq \frac{1}{2}$ as $T \geq 3$ by assumption.

With all these inequalities together, along with Lemma C.2 for the 1-Lipschitzness of the exploitability gap,

$$\begin{aligned}
\mathbb{E}\left[EG(w^T)^2\right] &\leq \mathbb{E}\left[\left(EG(w^{\tau,\star}) + \|w^T - w^{\tau,\star}\|_1\right)^2\right] \\
&\leq \mathbb{E}\left[\left(EG(w^{\tau,\star}) + \|w^{\tau,T} - w^{\tau,\star}\|_1 + \|w^T - w^{\tau,T}\|_1\right)^2\right] \\
&\leq 3\,\mathbb{E}\left[EG(w^{\tau,\star})^2 + \|w^{\tau,T} - w^{\tau,\star}\|_1^2 + \|w^T - w^{\tau,T}\|_1^2\right] \\
&\leq 3\left(\tau^2\left(\log(A) + \log(B)\right)^2 + \frac{4(A+B)}{\tau^2 T} + \frac{8}{T+1}\left(\log(A) + \log(B)\right)\right).
\end{aligned}$$

In particular, with $\tau^2 = \frac{2}{\log(A) + \log(B)}\sqrt{\frac{A+B}{T}}$, we obtain,

$$\mathbb{E}\left[EG(w^T)^2\right] \leq 3\left(4\sqrt{\frac{A+B}{T}} + \frac{8}{T}\right)\left(\log(A) + \log(B)\right)$$

hence, as $T \geq 4$ and $A + B \geq 4$,

$$\mathbb{E}\left[EG(w^T)^2\right] \leq 18\sqrt{\frac{A+B}{T}}\left(\log(A) + \log(B)\right)$$

and by taking the square root,

$$\|EG(w^T)\|_2 \leq 3\sqrt{2}\left(\frac{A+B}{T}\right)^{1/4}\sqrt{\log(A) + \log(B)}.$$

$\square$

**Theorem 8.1.** *Using Algorithm 3 with, for each $\mathcal{A}_i$, the algorithms and parameterization of Theorem 7.2, along with $T_i = 32(i2^i)$ and $S_i = 8(2^i)$, we obtain the bound*

$$\|EG(\mu^t, \nu^t)\|_2 \leq 30\left(\frac{A+B}{t}\right)^{1/4}\sqrt{\log(AB)\log(t)}$$

*for any total number of iterations $t = T_1 + \ldots + T_{i-1} + j$.*

*Proof.* Let $EG(i, s)$ be the exploitability gap of sub-algorithm $M_i$ after $s$ iterations. With the notations

$$C_{A,B} = 18\sqrt{A + B}\left(\log(A) + \log(B)\right),$$

we have from the proof of Theorem 7.2, taking $T = s$ and $\tau^2 = \frac{2}{\log(A)+\log(B)}\sqrt{\frac{A+B}{T_i}}$,

$$\mathbb{E}\left[EG(i, s)^2\right] \leq C_{A,B}\frac{\sqrt{T_i}}{s}$$

assuming $s \leq T_i$.

We will also use the notation, for any $j \leq T_i$, $s_i^j = \sum_{l=1}^{j} p_k^l$, the expected sum of calls of sub-algorithm $M_i$ during the loop $i$. Note that, because of the sampling method, the actual number of calls $s$ will be at least $s_i^j - 1$.

We notice, for any $\frac{1}{2}S_i\log(T_i) \leq j \leq S_i\log(T_i)$,

$$s_i^j = \frac{e^{\frac{j+1}{S_i}} - 1}{T_i\left(e^{\frac{1}{S_i}} - 1\right)}$$

and in particular,

$$
\begin{aligned}
s_i^j - 1 &= \frac{e^{\frac{j+1}{S_i}} - 1}{T_i\left(e^{\frac{1}{S_i}} - 1\right)} - 1 \\
&\geq \frac{4S_i}{5T_i}\left(e^{\frac{j+1}{S_i}} - 2\right) \\
&\geq \frac{4(1 - 2e^{-2})}{5}\frac{S_i}{T_i}e^{\frac{j+1}{S_i}} \\
&\geq \frac{S_i}{2T_i}e^{\frac{j+1}{S_i}} \quad (*)
\end{aligned}
$$

where we used $e^{\frac{1}{S_i}} \leq 1 + \frac{1}{S_i} + \frac{1}{S_i^2} \leq 1 + \frac{5}{4S_i}$, $e^{\frac{j+1}{S_i}} \geq e^{\frac{\log(T_i)}{2}} \geq e^2$.

Now, for any $j \leq T_i$ and $t = T_1 + ... + T_{i-1} + j$. We have

$$\mathbb{E}\left[EG(\mu^t, \nu^t)^2\right] = \underbrace{p_i^j\,\mathbb{E}\left[EG(i, s)^2\right]}_{\alpha(i,j)} + \underbrace{(1 - p_i^j)\,\mathbb{E}\left[EG(i - 1, s')^2\right]}_{\beta(i,j)}$$

where $s \geq s_k^u - 1$ and $s' \geq s_{i-1}^{T_{i-1}} - 1$.

**First term** $(\alpha(i,j))$:

- Either $j \leq \frac{S_i}{2}\log(T_k)$, then $\alpha(i,j) \leq p_i^j \leq \frac{1}{\sqrt{T_i}}$ using $EG(i, s) \leq 1$

- Either $\frac{1}{2}S_i\log(T_i) < j \leq S_i\log(T_j)$, in this case, using $(*)$ we obtain

$$s \geq s_i^j - 1 \geq \frac{S_i}{2T_i}e^{\frac{j+1}{S_i}} \geq \frac{S_i}{2}p_i^j$$

This yields

$$\alpha(i,j) \leq 2C_{A,B}\frac{\sqrt{T_i}}{S_i} = 8iC_{A,B}\frac{1}{\sqrt{T_i}}.$$

- Either $S_i\log(T_i) < j$, in this case, using $(*)$ again with $\lfloor S_i\log(T_i)\rfloor$ instead of $j$, we obtain:

$$s \geq s_i^j - 1 \geq s_i^j - 1 \geq \frac{S_i}{2T_i}e^{\frac{j+1}{S_i}} \geq \frac{S_i}{2}$$

which again yields:

$$\alpha(i,j) \leq 2C_{A,B}\frac{\sqrt{T_i}}{S_i} = 8iC_{A,B}\frac{1}{\sqrt{T_i}}.$$

as $p_i^j \leq 1$.

**Second term**  $(\beta(i,j))$

For the second term, which depends on the total number of call of algorithm $M_{i-1}$, we will lower-bound this number using only the calls during the loop $i-1$. As $T_{i-1} \geq \lfloor S_{i-1}\log(T_{i-1})\rfloor$, we have, using inequality $(*)$ on the loop $i-1$, with the previous quantity, as above,

$$s' \geq s_{i-1}^{T_{i-1}} - 1 \geq s_{i-1}^v - 1 \geq \frac{S_{i-1}}{2}$$

and we obtain:

$$\beta(i,j) \leq 2C_{A,B}\frac{\sqrt{T_{i-1}}}{S_{i-1}} = 8(i-1)C_{A,B}\frac{1}{\sqrt{T_{i-1}}}.$$

**Final bound**  : To conclude, we will also need the simple inequalities:

- $T_i \geq \sum_{l=1}^{i-1} T_l$ (and thus $t \leq 2T_i$ if $t$ is in loop $i$).

- $4T_{i-1} \geq T_i$

- $i \leq \log(T_{i-1})/\log(2)$

Using these, we obtain, for any $i$, $j$ and $t = T_1 + ... + T_{i-1} + j$:

$$\begin{aligned}
\mathbb{E}\left[EG(\mu^t,\nu^t)^2\right] &= \alpha(i,j) + \beta(i,j) \\
&\leq 8iC_{A,B}\frac{1}{\sqrt{T_i}} + 8(i-1)C_{A,B}\frac{1}{\sqrt{T_{i-1}}} \\
&\leq \frac{8}{\log(2)}\left(\sqrt{2} + 2\sqrt{2}\right)C_{A,B}\frac{\log(T_i)}{\sqrt{t}} \\
&\leq \frac{8}{\log(2)}\left(\sqrt{2} + 2\sqrt{2}\right)C_{A,B}\frac{\log(t)}{\sqrt{t}}.
\end{aligned}$$

Finally, taking the square root and using the definition of $C_{A,B}$,

$$\begin{aligned}
\|EG(\mu^t,\nu^t)\|_2 &\leq \sqrt{\frac{144}{\log(2)}\left(\sqrt{2}+2\sqrt{2}\right)}\left(\frac{A+B}{t}\right)^{1/4}\sqrt{(\log(A)+\log(B))\log(t)} \\
&\leq 30\left(\frac{A+B}{t}\right)^{1/4}\sqrt{(\log(A)+\log(B))\log(t)}.
\end{aligned}$$

$\square$

