# OpenReview forum: "The Harder Path: Last Iterate Convergence for Uncoupled Learning in Zero-Sum Games with Bandit Feedback"
_ICML.cc/2025/Conference — ICML 2025 poster_

### Official Review · Reviewer_TkNJ · 2025-03-09

**Overall Recommendation:** 2

**Summary:**

This paper improves the lower bound for learning matrix games with bandit feedback and last-iterate convergence in the uncoupled setting from $O(T^{-1/2})$ to $O(T^{-1/4})$. The authors then propose a black-box reduction from an algorithm with the so-called “output convergence” to the last-iterate convergence, though still requiring the computation of average policies. The authors then try to fix this drawback by proposing the regularized mirror descent algorithm and a variant of it by using the doubling trick.

**Claims And Evidence:**

I think some claims in this work might not be technically rigorous. Please see the weakness part below for details.

**Essential References Not Discussed:**

No.

**Experimental Designs Or Analyses:**

Not applicable.

**Methods And Evaluation Criteria:**

Yes.

**Other Comments Or Suggestions:**

Please see the weakness part above.

**Other Strengths And Weaknesses:**

**Weakness**
1. **Theoretical Results**: In the abstract, the authors claim that they propose two algorithms to achieve the optimal rate of $O(T^{-1/4})$. However, I think this is a bit misleading and even a bit overclaimed.
   * In Algorithm 1, if I understand correctly, the key insight is that an algorithm with “output convergence” can be translated into an algorithm with last-iterate convergence. Nevertheless, I think there is no fundamental difference between the definitions of “output convergence” and the average-iterate convergence. If an algorithm has the average-iterate convergence, then the output sequence $(\hat{\mu}^t, \hat{\nu}^t)$ can be chosen as $(\hat{\mu}^t, \hat{\nu}^t)=(\sum_{1\le i\le t} \mu_i, \sum_{1\le i\le t} \nu_i,)$, which is guaranteed to converge. This leads to at least two downsides of Algorithm 1 in this work: the two players are not fully uncoupled and it is still required to compute the average policy profile, which are also noted by the authors at the end of Section 6.
   * To tackle the above issues, the authors propose Algorithm 2, which is indeed fully uncoupled and does not require computing the average policy profile. However, its convergence guarantee is not anytime. To further fix this issue, the authors consider equipping their Algorithm 2 with the common doubling trick, at the cost of making the algorithm coupled again.

   Therefore, I have to say I do not think that the authors really establish an algorithm that has anytime last-iterate convergence and operates in a truly uncoupled manner, like the algorithm in [1]. Besides, [1] establish a high-probability convergence guarantee while the definition of the $\ell^p$ convergence in this work only permits the convergence in expectation.
2. **Presentation**: Some parts of this work seem to lack sufficient explanations. Lemma 6.1 is a key lemma in this work and I would suggest the authors give more discussions and the proof sketch (if possible) in the paper. Besides, on the RHS of Line 302-329, the authors just introduce the design of the “regularized mirror descent” but without giving any discussions or explanations about this. Why can this kind of design enable a convergence rate of $O(1/t)$ for the Bregman divergence? Further, what is the intuition behind the LHS of Line 336 and 338? Since $\mu^0$ is a uniform distribution, I think the operation on Line 336 is equivalent to the following FTRL-style update?
$$
\mu^t\in \arg\min_{\mu} –(1-\tau\eta^t)\langle \nabla \psi(\mu^{\tau,t}), \mu \rangle+ \psi(\mu)
$$

[1] Cai et al. Uncoupled and Convergent Learning in Two-Player Zero-Sum Markov Games with Bandit Feedback. NeurIPS, 2023.

**Questions For Authors:**

Please see the weakness part above.

**Relation To Broader Scientific Literature:**

The theoretical findings are new. But I am still concerned about its technical values. Please see the weakness part below for details.

**Theoretical Claims:**

I did not check the correctness of the proof details.

---

> ### Author Rebuttal · Authors · 2025-04-01
>
> We thank reviewer TkNJ for taking the time to review our article and for the clarity suggestion. We address the concerns below.
>
> >Nevertheless, I think there is no fundamental difference between the definitions of “output convergence” and the average-iterate convergence.
>
> We preferred to state the results and Lemma 6.1 under the assumption that $\mathcal{A}$ has a convergence of its output, whether it comes from averaging (as in the EXP3-IX algorithm that we use) or other means. We preferred to state it this way, as the output could also come from a reweighted average for example, or by sampling uniformly at random a policy among the ones played.
>
> >Therefore, I have to say I do not think that the authors really establish an algorithm that has anytime last-iterate convergence and operates in a truly uncoupled manner, like the algorithm in [1].
>
> While the anytime algorithms require a common seed and are not uncoupled in the same sense as in [1], they match the definition of uncoupledness that is required for the lower bound stated in Theorem 5.1 to hold. This lower bound relies on the absence of communication \textit{between} the iterations and does not exclude an agreement before the start of the iterations. One of the main points of the two anytime algorithms was especially to characterize the minimax rate.
>
> Furthermore, we would argue that the main weakness of Algorithm 2, the lack of anytime guarantees, is not problematic in practice. One of the main applications of last-iterate is to avoid the averaging of complex representation as explained in line 32-right, and the anytime convergence is not important for this matter.
>
> >Lemma 6.1 is a key lemma in this work and I would suggest the authors give more discussions and the proof sketch (if possible) in the paper.
>
> The paragraphs above Lemma 6.1 provide the key point behind the lemma and can be seen as a proof sketch, we will add more quantitative arguments and re-organize the section to make it more apparent.
>
> >Besides, on the RHS of Line 302-329, the authors just introduce the design of the “regularized mirror descent” but without giving any discussions or explanations about this. Why can this kind of design enable a convergence rate of $\mathcal{O}(1/T)$ for the Bregman divergence?
>
> We could add that the “regularized mirror descent” algorithm can roughly be seen as applying a regular mirror descent with the regularized operator $F_\tau$, hence the name, the two-steps update making the computation and the proof easier. This regularization makes the operator strongly monotone. Similarly to how strong convexity accelerates the convergence to the minimum, strong monotonicity makes the mirror descent convergent at a rate $\mathcal{O}(1/T)$. We propose to add a proof sketch of Lemma 7.1 to explain how we obtain this result.
>
> >Further, what is the intuition behind the LHS of Line 336 and 338? Since
>  $\mu^0$ is a uniform distribution, I think the operation on Line 336 is equivalent to the following FTRL-style update?
>
>  Indeed, the two updates are equivalent. However, the intuition between Line 336
>  $$\mu^{t} \gets argmin_{\mu_\in\Delta_A} (1-\tau\eta^t)KL(\mu,\mu^{\tau,t}) + \tau\eta^t KL(\mu,\mu^0)$$
>  is to regularize the iterates toward the uniform, to "stabilize" the algorithm and allow the convergence. This specific regularization is not normally present in FTRL and only appears in the above equation because of the $(1-\eta^t\tau)$ factor.

---

> > ### Comment · Reviewer_TkNJ · 2025-04-03
> >
> > I appreciate the detailed responses of the authors. Nevertheless, some of my key concerns remain unsolved:
> >
> > **Algorithm 1: “Output Convergence” and Average-iterate Convergence.**
> >
> > I agree that the definition of the “output convergence” considered in this work does not restrict on how the output convergent policy is generated (either by a reweighted average or "sampling uniformly at random a policy among the ones played"). Nevertheless, note that the original motivation of establishing algorithms with last-iterate convergence guarantees in the common literate is exactly to avoid *computing average policy* or *sampling a policy from the policy set* generated in the running of the players' algorithm, as this might induce additional (or even prohibitively large) computation or storage overhead. Therefore, I do not think there are fundamental differences between the definitions of “output convergence” and the average-iterate convergence, and my concern remains unsolved.
> >
> > **Algorithm 2 and 3.**
> >
> > * **Lacking anytime guarantees**: I also agree that lacking anytime guarantees - the downside of Algorithm 2 in this work - might not be a serious problem when implementing the algorithms in practice. Nonetheless, from the theoretical point of view, the improvement on the convergence rate of Algorithm 2 comes from more like at the cost of sacrificing some advantages of the algorithm in [1], which thus makes Algorithm 2 less appealing and somewhat limits the technical values of the algorithmic design and the results in this work.
> > * **Not truly uncoupled**: Similarly, I do concur that requiring a common seed between the two players (the downside of Algorithm 3 in this work) might also not be a critical problem that hinders the implementation of the algorithms in practice. However, again, this makes the improvement on the convergence rate of Algorithm 3 come from more like at the cost of sacrificing some good features of the algorithm in [1], and limits the technical values of the algorithmic design and the results to some extent.
> > * **Convergence in expectation**: Besides, I do not seem to find the responses by the authors to my concern that the algorithms in this work only have expected convergence, while the algorithm in [1] has a high probability convergence guarantee.
> >
> > Overall, I would definitely support the acceptance of this work, if there were no works such as [1]. However, currently, given the algorithm in [1], the aforementioned downsides of the algorithms in this work really prevent me from supporting the acceptance of this work, though the rates indeed have been improved in some sense.

---

> > > ### Author Response · Authors · 2025-04-07
> > >
> > > We again thank reviewer TkNJ for the response.
> > >
> > > >Algorithm 1: "Output convergence" vs. "Average-iterate convergence":
> > >
> > > It seems from your response that we agree on this matter. The output convergence assumption used in Algorithm 1 follows, in the case of EXP3-IX as in many other cases, from the average-iterate convergence of $\mathcal{A}$; the term "output convergence" is not used to hide this fact, but only to be slightly more general. This makes Algorithm 1 only relevant for matching the lower bound, and is the issue stated on Lines 295-300 (right):
> > >
> > > "One of the main points of the last-iterate convergence is to avoid the computation of the average necessary in the regret-based algorithm. Not only is this computation still required here (the output of EXP3-IX is an average), but also needs to be done at almost every iteration."
> > >
> > > >Algorithm 2 and 3:
> > >
> > > We understand your concerns about the lack of anytime guarantees of Algorithm 2 and the common seed of Algorithm 3 as we state in the papers, as we do not strictly improve the results of [1]. Nonetheless, these algorithms are the only ones to match up to logarithmic factors the $\mathcal{O}(t^{-1/4})$ lower bound and characterize the optimal rate, which alone makes the results interesting in our opinion.
> > >
> > > >Convergence in expectation
> > >
> > > We apologize for not responding to this part, as we saw it as a statement of a weakness mentioned for example in Table 1 of the paper, which we therefore agree with. Note that, as pointed out by Reviewer eajt, [1] provides in Appendix C a rate of $\mathcal{O}(t^{-1/6})$ in expectation, which we will add to this table.

---

### Official Review · Reviewer_eajt · 2025-03-11

**Overall Recommendation:** 4

**Summary:**

This paper studies the last-iterate convergence rates of uncoupled learning dynamics in two-player zero-sum games with bandit feedback. One of the main contributions of the paper are lower bounds for uncoupled learning dynamics: (1) $\Omega(t^{-1/(2+p)})$ lower bound for any-time $\ell_p$ last-iterate convergence rates when $p \in (0,2]$; and (2) $\Omega(t^{-1/4})$ lower bound for $p \ge 2$. Then the authors propose two algorithms for the problem. The first one is a general framework that could transform the guarantee of output sequence to last-iterate convergence by simultaneously exploring or exploiting. By appling EXP3-IX, this framework gives $\tilde{O}(t^{-1/(2+p})$ last-iterate convergence rates. The drawback of this approach is (1) it is not fully uncoupled as it requires shared randomness; (2) it still requires iterate-averaging in execution. The second algorithm uses regularization and achieves $O(t^{-1/4})$ $\ell_2$ last-iterate convergence. However, the convergence is not any-time as the time horizon is required to choose the step size. A doubling trick fixes this issue but introduces synchronization again just like the first algorithm.

**Claims And Evidence:**

Yes. The claims are supported by proofs.

**Essential References Not Discussed:**

I think related works have been substantially discussed. The only comment I have is regarding the results in [1]. For uncoupled learning in zero-sum games, there are two results in [1] for last-iterate convergence rates measured by the duality gap: (1) $O(t^{-1/8})$ high-probability bound and (2) $O(t^{-1/6})$ bound in expectation. This paper discusses the first one but not the second one. Since the current paper focuses on last-iterate convergence in expectation, comparing the second bound in [1] would make the discussion more complete.

**Experimental Designs Or Analyses:**

N/A

**Methods And Evaluation Criteria:**

N/A

**Other Comments Or Suggestions:**

Some equations may be explained in more detail. See my question.

**Other Strengths And Weaknesses:**

This is a good paper. The presentation is clear and easy to follow, and the authors thoroughly explain the high-level ideas.

This paper provides the first lower bounds for uncoupled learning under bandit feedback and separate bandit feedback with full gradient feedback. These results are very interesting.

The algorithms proposed in the paper are also interesting. In particular, I appreciate that the authors distinguish "any-time" last-iterate convergence from convergence when the time horizon $T$ is known. Clearly, the former is stronger and the "real" guarantee one seeks for last-iterate convergence. The authors do a good job of clearly stating the weaknesses of both of their algorithms, which I appreciate. As the authors admit, both algorithms fail to achieve the any-time last-iterate convergence in a fully uncoupled way, yet they could yield $O(t^{-1/4})$ convergence if we allow some shared randomness.

**Questions For Authors:**

1. Could you explain how to derive line 682 from line 679-680? Maybe it is trivial, but I think some explanation would be helpful for the readers.

**Relation To Broader Scientific Literature:**

This paper studies the last-iterate convergence rates of uncoupled learning dynamics in two-player zero-sum games with bandit feedback. The paper's results contribute to the broad line of research in learning in games, which has recently been applied in machine learning. More specifically, uncoupled learning dynamics in games have been studied recently: [1] gives the first algorithm with $O(t^{1/8})$ high-probability last-iterate convergence and $O(t^{-1/6})$ expected last-iterate convergence; [2] generalizes this result to monotone games. The current paper provides the first lower bounds for the problem. The paper also offers new algorithms with improved convergence rates. Although strictly speaking, they are not fully uncoupled,

[1] Cai, Yang, Haipeng Luo, Chen-Yu Wei, and Weiqiang Zheng. "Uncoupled and convergent learning in two-player zero-sum markov games with bandit feedback." NeurIPS 2023

[2] Dong, Jing, Baoxiang Wang, and Yaoliang Yu. "Uncoupled and Convergent Learning in Monotone Games under Bandit Feedback." arxiv preprint, 2024

**Theoretical Claims:**

I checked the proofs for the lower bounds and the other proofs also look fine to me.

---

> ### Author Rebuttal · Authors · 2025-04-01
>
> We thank Reviewer eajt for the review and for pointing out a result that we missed in one of our references.
>
> >I think related works have been substantially discussed. The only comment I have is regarding the results in [1]. For uncoupled learning in zero-sum games, there are two results in [1] for last-iterate convergence rates measured by the duality gap: (1) $\mathcal{O}(T^{-1/8})$ high-probability bound and (2) $\mathcal{O}(T^{-1/6})$ bound in expectation. This paper discusses the first one but not the second one. Since the current paper focuses on last-iterate convergence in expectation, comparing the second bound in [1] would make the discussion more complete.
>
>  Indeed, this $\mathcal{O}(t^{-1/6})$ result in expectation is an unhighlighted result of [1] that we missed and that should appear in Table 1. Their result is stated for the $\ell^1$ norm in Theorem 4 of Appendix $C$, but a slight change to the proof seems to make it hold for the $\ell^2$ norm.
>
> >Could you explain how to derive line 682 from line 679-680? Maybe it is trivial, but I think some explanation would be helpful for the readers.
>
> The link between the two lines is relatively direct computation-wise, but it is far from obvious and deserves more explanation. The internal randomness $\omega$ of the two players is independent of the game by construction, which implies that $\mathcal{D}(P_0^{\omega},P_{\theta^T}^{\omega})$ is $0$. The same can be said of $a^t|z^{t-1}$ for any $t$: the policy $\mu^t$ of the min-player is predictable with respect to $(\sigma(z^t))$, and the action $a^t$ is sampled from $\mu^t$ independently from the past. This implies that $\mathcal{D}((P_0^{a^t|z^{t-1}},P_{\theta^T}^{a^t|z^{t-1}})$ is also $0$ for all $t$.
>
> Finally, the terms of the sum come from developing the observation of the min-player given the predictable policy $(1/2+\delta^t,1/2-\delta^t)$ of the opponent, as explained in line 189-right of the main paper.
> These explanations will be added to the proof.

---

### Official Review · Reviewer_U4sa · 2025-03-19

**Overall Recommendation:** 3

**Summary:**

The paper aims to improve the upper and lower bounds in  Last Iterate Convergence for Uncoupled Learning in Zero-Sum Games with Bandit Feedback

**Claims And Evidence:**

The evidence is fairly clear, but I will provide a detailed explanation of my questions in the following sections.

**Essential References Not Discussed:**

I would like to request a comment and a more detailed explanation regarding lines 195–207.

Specifically:
	1.	Is there any existing literature on the concept of passing bits to others through artificial choices? If so, I would appreciate references or further discussion on this point.
	2.	I would also like to better understand an aspect related to the notion of uncoupled dynamics. Based on the authors’ description, it is unclear whether the step size is a common constant step size, a varying common step size, or if different step sizes have been collaboratively chosen to shape the dynamics—or if they truly remain “uncoupled.”
	3.	Why exactly do we want an uncoupled algorithm rather than the even more general notion of independent play? Does this choice primarily aid parallelization, or does it serve as a better model of real-world scenarios?

**Experimental Designs Or Analyses:**

It is an excellent paper that does not require experiments.

However, what I would like to see clarified is whether the paper models the behavior of a game-learning process between two independent agents or if it simply describes a collaborative uncoupled dynamics algorithm that will be employed in one server.

**Methods And Evaluation Criteria:**

N/A

**Other Comments Or Suggestions:**

Clarification Requests & Suggested Revisions

Page 19, Line 1038
	•	Why is there a norm applied to the KL-divergence? Could you clarify the reasoning behind this?

Page 18
	•	In the secondary order bound, please clarify whether the negative third-order term pertains specifically to exponential gradient descent (for the case of the entropy regularizer).
	•	Initially, I attempted to prove this for any regularizer, but I do not believe it holds for \ell_2 regularization. Am I mistaken?
	•	Are there other regularizers (e.g., Tsallis) that satisfy the property \nabla h^3 < 0?
	•	If the claim is indeed correct for all regularizers, please explain why.

Page 17
	•	I would appreciate a more detailed explanation of why the property \equiv holds in this context.
	•	Personally, I prefer to express mirror descent using the constrained optimality condition inequality. While this does not change the correctness of the algorithm, I would appreciate it if you could state a lemma indicating whether this applies only to entropy regularization or to all regularizers.

Page 16
	•	This part was quite confusing to me. In Page 16, it is evident that the authors follow a constant-sum formulation rather than zero-sum. While this does not impact the dynamics, I would like to understand:
	•	(a) Why was this choice made?
	•	(b) Why does F_\tau have 1-L for the y-player, whereas the F operator has -L? These two seem inconsistent—could you clarify?

Page 15
	•	In the last three inequalities, I believe that we should have B_t \log B in the pre-last summand (first line).
	•	I would recommend multiplying all summands by 2, and for the first one, multiplying by \sqrt{t} (I believe this might be a typo).

Proofreading Issues & Formatting Corrections
	•	Line 797: Missing closing parenthesis.
	•	Line 650: Should be |\delta^t/3| instead of the current notation.
	•	Line 640: There are two colons instead of one.

Page 13
	•	The paper uses the additive KL divergence property. Could you please rewrite this section?
	•	The issue is that \mathbb{E}_0 does not refer to P_0, but rather to P_0^{z^T}.
	•	As a result, when summing over the history, there are redundant elements.
	•	I am not suggesting that it is incorrect, but I would appreciate a detailed response in your replies to verify that there are no proofreading issues here.

Line 689
	•	Please include one explanatory sentence for the reader, clarifying which terms are zero and why in this line.

⸻

Final Notes

I appreciate the effort that has gone into this paper. The above points focus on improving clarity, correctness, and presentation to ensure that all arguments are fully transparent and well-supported.

Would you be able to address these concerns in your response?

**Other Strengths And Weaknesses:**

For me, the most interesting aspects of this paper are:

(a) The honesty in the statements. I truly commend one or more of the authors for their transparency in presenting the results.

(b) The structured distinction between different iterate types. While it may not necessarily be the most accurate modeling approach, I appreciate the authors’ effort to clearly differentiate between last-iterate, average-iterate, output-iterate, and how these relate to asymptotic vs. general results.

As a faculty member, if all the papers I reviewed were of this level of quality, I would genuinely enjoy reviewing even more.

**Questions For Authors:**

Questions and Clarifications for Discussion

I would greatly appreciate it if you could provide responses to the questions I have posed in the previous sections.

One of the most critical aspects of the paper is the discussion surrounding learning-output-last-iterate iteration. Having served on multiple review committees, I have often encountered disputes between different models and interpretations of these notions.

Let’s see if I have understood correctly:

	•	A sequence is called learning if its past completely determines its predictability.

	•	Consider Rock-Paper-Scissors as an example. We say that a sequence is learning if its behavior depends on observations from previous rounds.

	•	If the distribution is stationary, then it is also a learning sequence, correct?

	•	In other words, any sequence that can be computed based on past information qualifies as a
learning sequence.

Thus, the natural question arises:

##What constitutes a non-learning sequence?##

Let’s now examine a classic question that often arises in the algorithms community:

	•	Suppose I compute a last-iterate sequence of extra-gradient updates inside a subroutine.

	•	However, in the main function, I output the time-averaged sequence as my “actual”
sequence.

If I understood correctly, the paper attempts to reconcile this distinction using the output and last-iterate sequence framework.
	•	That is, \hat{\mu}_t represents the averaged sequence, while \mu_t represents the last-iterate sequence.

	•	Is this the correct intuition?

Defining the $played$ Distribution in a Bandit Setting (lines 192)

What exactly determines the distribution being $played$?
	•	In a bandit setting, how do we rigorously define this distribution?

	•	Unlike in a centralized version of bandit game, we do not submit our mixed strategy to an authority that then returns a bandit feedback element, right .

	•	Instead, in a bandit setting, we simply announce pure strategies.

Could you clarify these details? I appreciate the effort you have made to formalize these distinctions—even if we later find issues with the model during the rebuttal discussion, your approach to structuring these ideas is commendable.

⸻

Final Request: Intuition Behind the Doubling Modified Trick

While I was able to follow and learn a lot from this paper, I found the method for the doubling modified trick highly unintuitive.

Could you provide a more detailed intuition behind:
	1.	How this method was chosen (designed)?

	2.	Why it works?

This was the part of the analysis that had the least intuition regarding the computations, and I would greatly appreciate further clarification.


P.S. A satisfactory response to the questions I’ve raised—especially regarding the distinctions from prior work, the modeling clarity around bandit feedback and iterate definitions, and the theoretical subtleties—could lead me to raise my score to “Strong Accept”.

P.S.2 I may have asked this already, but I was genuinely surprised that the authors do not require any form of \epsilon-greedy exploration.
One possible explanation is the clever anchoring step involving the term
$(\tau\cdot \eta_t) D_{\mathrm{KL}}(m, m_0)$,
but I’d really like to understand how the analysis successfully avoids the need for exploration noise.
	•	Is this avoidance enabled by the negative third derivative of the regularizer (e.g., $\nabla^3 h < 0$)?
	•	Or is there another mechanism or structural property of the setup that removes the need for explicit exploration? When is the assumption of the extrapolation necessary and when not?

A short clarification on this point would be greatly appreciated, as it’s quite non-standard and technically impressive—if valid.

**Relation To Broader Scientific Literature:**

The paper achieves better learning rates than those found in the literature.

Without having reviewed the results of Cai and Dong on zero-sum Markov games, could you explain how their baseline translates to the stage-game setting? The reason I ask is that Markovian noise introduces additional stochasticity, and I would like to determine whether this improved learning rate can be obtained “for free” in this setting. Finally, I would also like to know whether the authors believe a similar result could be achieved using standard gradient descent or optimistic gradient descent under bandit feedback, rather than requiring full feedback.

**Theoretical Claims:**

As I mention later in my review, I would like to congratulate the authors on their honest and transparent presentation of results. With that in mind, I have a few questions to ensure we are on the same page:
	1.	Why are we attempting to minimize communication if we allow the loss function to be designed with a common regularization term (with a shared regularizer weight)? Could these two characteristics be relaxed simultaneously?
	2.	While I conducted a careful proof-reading, I did not have time to fully grasp the necessity of the shared random variable seed in communication. Why can’t the incoming p^i play the role of u? I am trying to understand why we cannot instead leverage arbitrary information from strategies in the sequence.
	3.	In lines 236–243, there is a reference to an information-theoretic old-fashioned result. Could you elaborate on this? Specifically, why does allowing observation enable us to estimate each matrix entry? I assume the matrix size is at least O(t^{-1/2})—could you provide a more detailed explanation? I found this particularly interesting.
	4.	One aspect that initially confused me was the lower bound. In the proof in the Appendix, you exclusively examine three different games, rather than a random matrix with Bernoulli entries. What exactly is meant by “Bernoulli” in the main text? Was this an earlier idea, or am I missing something in the problem formulation?

---

> ### Author Rebuttal · Authors · 2025-04-01
>
> We would like to express our sincere gratitude to Reviewer U4sa for their very comprehensive and constructive feedback. We appreciate the encouragement, particularly the acknowledgement of our "honest and transparent presentation of results".
>
> We prepared a response for all of the questions, but we were greatly limited by the 5000 characters limit. We will only be able to submit (once) the response to the rest of the questions (especially some of the longest replies) after the reviewer's response.
>
> >Q1
>
> For our analysis to hold, the regularization term currently needs to be the same for both players for Algorithm 2. As it is a hyper-parameter, we consider this requirement to be substantially weaker than actual communication.
>
> >Q2
>
> The reason we use this shared seed that allows the sampling of a shared Bernoulli with parameter $p^i$ at each iteration, and not a Bernoulli directly is to make sure the two players have exactly the same sample $B^i$. The two players indeed need to explore and exploit simultaneously, as the exploitation (i.e. playing the output of $\mathcal{A}$) of one would otherwise bias the exploration of the other: the guarantees of $\mathcal{A}$ only holds if both players play the algorithm over the relevant iterations.
>
> >Q3
>
> If at each iteration $i$, the actions $a^i$ and $b^i$ are observed by the two players, then, for any entry $(a,b)$ of the matrix, the expected value of this entry can be estimated by averaging all of the observations for which $(a^i,b^i)=(a,b)$. Assuming that each action profile is played on average $\Omega(t)$ times over $t$ iterations as $t$ scales to $+\infty$, then each entry of the matrix can be estimated with a precision of $\tilde{\mathcal{O}}(t^{-1/2})$ using Hoeffding inequality. Suppose each player computes its minimax strategy associated with the resulting estimated matrix (which can be done "offline", without observations) and plays it at each iteration. It can then be shown that the exploitability gap scales with the precision of $\tilde{\mathcal{O}}(t^{-1/2})$.
>
> >Q4
>
> As explained in line 620, the proof also uses random matrices with Bernoulli entries, the entries of the matrix $M^\epsilon$ being the parameter of these Bernoulli. The confusion might come from lines 647-669, where we do the computation directly using this matrix, as the exploitability gap is defined for the expected loss matrix. We will add a sentence to clarify.
>
> > Markov games question
>
> We are not sure we fully understand the question. We assume that "learning rate" refers to our guarantees. While their second algorithm reduced to a single stage seems to be similar to their first algorithm on matrix games, the opposite transformation (transforming an algorithm that works on matrices into an algorithm that works on Markov games) is not trivial. In particular, their approach relies on estimating the value associated with each state.
>
> >gradient descent question
>
> Any mirror descent algorithm (including gradient descent) would not converge in general. In particular, if a fully mixed equilibrium exists, then the divergence between this equilibrium and the iterates is non-decreasing on expectation at each iteration.
> We think optimistic gradient descent would not work, but we think optimistic multiplicative weights update (which relies on the Shannon entropy instead) could if we use two different rates for the intermediate iterates and the actual iterates. We are not sure whether the optimal rate of $\mathcal{O}(t^{-1/4})$ is attainable with this method and if an additional problem-dependent constant would be necessary.
>
> >page 19
>
> Typo
>
> >page 18
>
> Indeed, this property does not hold for any regularizer, although it works for the most common ones, such as Shannon, Tsallis or the log-barrier (we only get $\nabla h^3 \leq 0$ for the $\ell_2$ regularizer).
>
> >page 17
>
> It follows the fact that the gradient of the minimizer of a convex function belongs to the opposite of the (here constant) normal cone associated with this point and the constraints.
>
> >page 16
>
> (a) This choice was made as Lemma C.1 relies on the estimated loss being non-negative.
> (b) typo
>
> >page 13
>
> Each term $t$ is the expectation of the same KL divergence with $\theta^{t-1}$ taken under $P_0^{z^{t-1}}$. As $z^{t-1}$ is the only random variable in each term of the sum, this is the same as taking the expectation under $P_0$ directly.
>
> >line 689
>
> See reviewer 3.
>
> > Questions for authors
>
> While a learning sequence is predictable using the internal randomess and the past actions and reward, a non-learning sequence could be based on the unknown entries of the matrix $L$, for example, which would go against the lower bound.
>
> The next intuition is correct
>
> In this setting, it is easier to consider that we announce our mixed strategy to an authority, which then returns both the sampled action and the associated reward. Otherwise, the convergence would indeed fail with the pure strategies.

---

> > ### Comment · Reviewer_U4sa · 2025-04-01
> >
> > I may have asked this already, but I was genuinely surprised that the authors do not require any form of \epsilon-greedy exploration. One possible explanation is the clever anchoring step involving the term $(\tau\cdot \eta_t) D_{\mathrm{KL}}(m, m_0)$, but I’d really like to understand how the analysis successfully avoids the need for exploration noise.
> >
> > I would appreciate a more mathematically precise clarification regarding the avoidance of greedy exploration

---

> > > ### Author Response · Authors · 2025-04-03
> > >
> > > There is again a hard limit of one response with 5000 characters. We include below the responses that were missing from the rebuttal, as it was not possible to give a complete answer to all of the questions within this limit. Despite this issue, we thank once more Reviewer U4sa for asking many relevant questions.
> > >
> > > >Greedy exploration
> > >
> > > The algorithm relies on an estimate of the loss, based on importance sampling. As this estimator is unbiased, it only appears through the "variance" term $\mathcal{D}(w^t,\tilde{w}^{\tau,t+1})$ that appears in the analysis (Lemma 7.1), which can be shown to be in expectation at most
> > > $$(\eta^t)^2\sum_{i=1}^{A+B} (w^t_i\nabla^2 h(\overline{w}^t_i))^{-1}$$
> > > using Taylor, for some $\overline{w}^t_i$ between $w_i$ and the unprojected $\tilde{w}^{t,\tau}_i$. With a gradient descent (relying on the $\ell^2$ regularization), $\nabla^2 h$ is constant equal to $1$, and this term is not bounded as $w_i^t$ approaches zero. Some $\epsilon$-greedy exploration would limit this term to $(A+B)(\eta^t)^2/\epsilon$, but degrade the last-iterate guarantees.
> > >
> > > Instead, we rely on a regularization that is more suited to the problem. The Shannon entropy satisfies both $\nabla^2 h(w_i^t)=1/w_i^t$ and, as you mentioned, $\nabla^3 h\leq 0$, which guarantees using the non-positivity of the estimate of the opposite of the loss and the definition of $\tilde{w}_i^{t,\tau}$ that $\nabla^2 h(\overline{w}_i^t)\geq 1/w_i^t$. The $w$ dependence in the upper bound of the divergence then disappears. Note that this works for any regularizer that satisfies $\nabla^2 h(w_i)=\Omega(1/w_i)$ and $\nabla^3 \leq 0$, this also includes Tsallis entropy and the log-barrier.
> > >
> > > >Doubling trick
> > >
> > > The main idea behind this method is that, in contrast to the regret setting, using a doubling trick with a hard reset is not possible because of the last iterate constraints. For this reason, instead of suddenly switching to the new instance with a reduced regularization, the algorithm keeps the previous last iterate and plays a mix between the former and the new instance. It is then able to progress with the smaller regularization while still playing a good policy on average, thanks to this former best iterate being played with an initial way higher probability. As the new instance progresses, we are able to decrease this probability and thus focus on the new (asymptotically better) instance.
> > >
> > > >Passing bits
> > >
> > > We are not aware of references on this point in this precise context. In a multiplayer multi-armed bandit problem in which collision occurs if two players choose the same arm, [1] proposes a protocol of communication between the agents relying on these collisions.
> > > One method could be the following, for player 1 to pass a bit of information to player 2:
> > >
> > > -A parameter L is fixed to be the same for both players.
> > >
> > > -To pass a $1$, player 1 plays the first action $L$ times, then the second $L$ times, and so on. To pass a $0$, he plays only the first action $A\times L$.
> > >
> > > -The second player plays its action uniformly and keeps track of the losses of each batch of $L$ iterations separately. If for a given action, the distribution of losses is statistically different between two batches, the output is 1, otherwise, it is 0.
> > >
> > > >Step size
> > >
> > > For simplicity of the analysis and of the presentation, a common step size has been chosen, but this is not mandatory: the two players can take different step sizes, and the algorithm will enjoy the same guarantees up to some constant factor. The only requirement for the rate to remain the same is that the step size is higher than $1/(t\tau)$ asymptotically for any player, as otherwise, the regularization is too strong and prevents any substantial learning.
> > >
> > > >Uncoupled algorithm
> > >
> > > One of the main reasons behind the study of uncoupled algorithms is that it indeed better models the way a practical algorithm would learn an actual game. We think that learning how to play the game using the feedback of a profile of actions, rather than with each action independently, is fundamentally slower as the size/complexity of the game increases, as it amounts to learning a matrix (of size $A\times B$) rather than learning two vectors (of size $A+B$).
> > >
> > > [1] Etienne Boursier, Vianney Perchet, SIC-MMAB: Synchronisation Involves Communication in Multiplayer Multi-Armed Bandits, NeurIPS 2019

---

### Official Review · Reviewer_bHVB · 2025-03-23

**Overall Recommendation:** 3

**Summary:**

This paper studies a zero-sum matrix game in which two players repeatedly select stochastic policies, sample actions, and receive stochastic losses—without access to the underlying payoff matrix—that depend on their joint actions. The authors aim to develop an uncoupled algorithm that independently controls each player without explicitly observing the other player’s actions, while ensuring last-iterate convergence of the players' policies to a Nash equilibrium.
The paper establishes a convergence rate lower bound of $\Omega(T^{-1/4})$ for this problem. To address the challenge, the authors propose two algorithms (Algorithm 2 and Algorithm 3). They show that Algorithm 2 achieves an $\ell^2$ convergence rate of $O(T^{-1/4})$, while Algorithm 3 achieves an $\ell^2$ convergence rate of $\tilde{O}(T^{-1/4})$.

**Claims And Evidence:**

It is concerning that the abstract claims both proposed algorithms achieve the optimal convergence rate. Could the authors kindly clarify whether the algorithms match the lower bound up to constant factors, only in terms of the order of $T$, or if the rates differ by logarithmic factors (e.g., $\log(T)$)?

**Essential References Not Discussed:**

Could the authors kindly provide appropriate citations for the transformation procedure described in Section 6?

**Experimental Designs Or Analyses:**

This paper does not provide any numerical study.

**Methods And Evaluation Criteria:**

The evaluation criteria (convergence rates) make sense.

**Other Comments Or Suggestions:**

The notation $\ell^p$ ($\ell^t$) appears to be used with different meanings in various parts of the paper, which may lead to confusion. It would be helpful if the authors could clarify or standardize the notation to improve readability.

**Other Strengths And Weaknesses:**

Strengths:
- The paper addresses a complex setting of zero-sum games by incorporating bandit feedback, uncoupled learning, and last-iterate convergence.
- It provides a convergence rate lower bound for the considered problem, contributing to the theoretical understanding of this setting.

Weakness:
- The presentation of the theoretical results is unclear.
- This paper does not include any numerical validation to support the effectiveness of the proposed algorithms.

**Questions For Authors:**

- Could the authors confirm whether, to the best of their knowledge, the convergence rate upper and lower bounds presented in this work are the tightest currently known?
- If so, could the authors kindly elaborate on the analytical tools or techniques they employed to derive a tighter lower bound on the convergence rate compared to prior works?

**Relation To Broader Scientific Literature:**

This paper contributes to the broader online learning research community by providing a tighter (compared to prior works, e.g., [Cai et al., 2023]) convergence lower bound for a specific setting of zero-sum matrix game.

**Theoretical Claims:**

The consistent use of Big-$O$ notation to express lower bounds throughout the paper is somewhat confusing, as Big-$\Omega$ notation is typically used for lower bounds. For instance, this appears in Section 1 when referencing the lower bound from [Cai et al., 2023], as well as in Table 1 and Section 2. Could the authors kindly clarify whether this usage is intentional?

---

> ### Author Rebuttal · Authors · 2025-04-01
>
> We thank Reviewer bHVB for taking the time to review our submission and especially for pointing out the issues in the notations. We address the concerns below.
>
> >It is concerning that the abstract claims both proposed algorithms achieve the optimal convergence rate. Could the authors kindly clarify whether the algorithms match the lower bound up to constant factors, only in terms of the order of $T$ or if the rates differ by logarithmic factors (e.g., $\log (T)$)?
>
> The rate is indeed only matched up to logarithmic factors, and we do not try to hide it in the paper. In the abstract, this comes from an oversight due to the change of the $\tilde{\mathcal{O}}$ notation to $\mathcal{O}$ as the lower bound does not include any log factor. We will change the abstract to address this confusion.
>
> >The consistent use of Big-$\mathcal{O}$ notation to express lower bounds throughout the paper is somewhat confusing, as Big-$\Omega$ notation is typically used for lower bounds. For instance, this appears in Section 1 when referencing the lower bound from [Cai et al., 2023], as well as in Table 1 and Section 2. Could the authors kindly clarify whether this usage is intentional?
>
>  This is an oversight, and we will change the relevant Big-$\mathcal{O}$ notations into Big-$\Omega$ (including in the abstract).
>
> >Could the authors kindly provide appropriate citations for the transformation procedure described in Section 6?
>
> While the idea behind the transformation procedure is natural, we are not aware of an article using this trick to transform an average profile convergence into a last-iterate convergence. If you have any precise references in mind that we did not stumble upon, please do not hesitate to give them so that we can add them to the paper
>
> >The notation $\ell^p$ ($\ell^t$) appears to be used with different meanings in various parts of the paper, which may lead to confusion. It would be helpful if the authors could clarify or standardize the notation to improve readability.
>
> Indeed, we use the notation $\ell^p$ for the convergence of the sequence and $\ell^t$ for the loss, and we assumed changing one of the two would bring more confusion. As the two refer to two completely different objects, we considered this overlap of notation acceptable. However, we propose to instead talk of the convergence of the sequence of the random variables $EG(\mu^t,\nu^t)$ in the $L^p$ space toward $0$, which we also believe to be better in addition of avoiding this overlap.
>
> >Could the authors confirm whether, to the best of their knowledge, the convergence rate upper and lower bounds presented in this work are the tightest currently known?
>
> The rates presented in Table 1 are the tightest currently known for high probability convergence. There also exists a $\mathcal{O}(t^{-1/6})$ rate for the $\ell^2$ convergence that was pointed to us that we will add to the table (and which we directly improve).
>
> >If so, could the authors kindly elaborate on the analytical tools or techniques they employed to derive a tighter lower bound on the convergence rate compared to prior works?
>
> The idea of the tighter lower bound is the focus of section 5 and is based on improving the classical lower bound on best arm identification (see e.g. [1]) for our context. This lower bound is based, from the point of view of one player, on the difficulty between distinguishing a sequence of $T$ Bernoulli $\mathcal{B}(1/2)$ and a sequence of $T$ Bernoulli $\mathcal{B}(1/2-\varepsilon)$, which is required to guarantee $\epsilon$-optimality. We show that $\epsilon$-optimality in the context of last-iterate convergence for games can require distinguishing between a sequence of $T$ Bernoulli $\mathcal{B}(1/2)$ and a sequence of $T$ Bernoulli $\mathcal{B}(1/2-\varepsilon_t)$ for $t$ varying between $1$ and $T$, with $\varepsilon_t=\mathcal{O} (EG(\mu^t,\nu^t))$. This implies a trade-off at each iteration between getting more information on the $\varepsilon$-optimal strategy and playing near-optimal profiles $(\mu^t,\nu^t)$, which explains the worse rate.
>
> [1] Jean-Yves Audibert, Sébastien Bubeck. Best Arm Identification in Multi-Armed Bandits, COLT 2010

---

> > ### Comment · Reviewer_bHVB · 2025-04-02
> >
> > I really appreciate the authors' clarification in response to my questions. I encourage them to proceed with implementing the changes discussed in the rebuttal. Given the commitment to these changes, I am raising my overall recommendation score.

---

> > > ### Author Response · Authors · 2025-04-07
> > >
> > > We thank Reviewer bHVB again for taking the time to read our response and for improving his recommendation score.

---

### Decision · Program_Chairs · 2025-05-01

**Decision:**

Accept (poster)

**Comment:**

This paper establishes a T^{-1/4} lower bound for the last-iterate convergence rate in decoupled two-player zero-sum games, improving upon the previously known T^{-1/2} bound. The lower bound construction is simple and interesting. The authors also present two algorithms that achieve matching upper bounds. As Reviewer TkNJ and eajt pointed out and mentioned in the paper, these algorithms require strong coordination between the players such as shared randomness or synchronization, so it's not as natural as previous algorithms. Nevertheless, they still satisfy the definition of decoupling. Overall, the lower bound offers a meaningful contribution to our understanding of this problem, and I think the paper could be accepted.